# THE COST OF PRIVACY IN FAIR MACHINE LEARNING

## ABSTRACT

A common task in fair machine learning is training ML models that preserve certain summary statistics across subpopulations defined by sensitive attributes. However, access to such sensitive attributes in training data is restricted and the learner must rely on noisy proxies for the sensitive attributes. In this paper, we study the effect of a privacy mechanism that obfuscates the sensitive attributes from the learner on the fairness of the resulting classifier. We show that the cost of privacy in fair ML is a decline in the generalizability of fairness constraints.

## 1 INTRODUCTION

The fairness of machine learning systems is gaining increasing attention in recent years. Among the numerous fairness objectives is ensuring that a machine learning model does not discriminate against subpopulations that are typically identified by sensitive attributes (*e.g.*, race, gender). When training a fair model and evaluating model bias, it is necessary to possess sensitive attributes; however, access to and use of such sensitive data is frequently prohibited by laws and regulations. Credit card companies, for instance, are not permitted to inquire about a person's race when they apply for credit, yet they must demonstrate that their decisions are not discriminatory (Chen et al., 2019).

Ideally, sensitive personal information should not be disclosed during the training of ML models. However, it is impossible to ensure exact notions of fairness (such as demographic parity or equality of opportunity) without any knowledge of the sensitive data. Fortunately, differential privacy (Dwork et al., 2006) is a promising workaround, which can offer a graceful compromise between privacy and utility. Mozannar et al. (2020) propose to release sensitive attributes in a locally differentially private way: adding noise to the sensitive data so that adversaries cannot infer any information with high confidence about a single record.

The advantage of the privacy mechanism proposed by Mozannar et al. (2020) is an invariance property: exact notions of fairness with regard to true sensitive attributes and noisy sensitive attributes are equivalent. An implication of the invariance property is that the optimal model of fairness can be learned at the population level. Nonetheless, it remains unclear what the precise statistical impact of privacy on fairness is.

In this work, we study the statistical cost of privacy on fairness in the task of learning fair ML models with differentially private sensitive attributes. The main benefits of the developed theory are

1. **statistically principled:** We propose a statistically principled metric to characterize the cost of privacy on fairness. A restricted notion of statistical efficiency *precisely quantifies* the privacy cost asymptotically.
2. **interpretable:** Privacy leads to a decline in the statistical efficiency. Such efficiency loss is interpretable: it explicitly depends on the privacy budget, the subpopulation imbalance level, and few other problem-specific parameters.

The rest of this paper is organized as follows. In Section 2, we formalize the problem setup, which consists of the constrained stochastic optimization problem for fair machine learning, the local differential privacy mechanism for releasing sensitive attributes, the learning procedure of fair model using private sensitive attributes, and the definition of asymptotic relative efficiency in terms of fairness violations. In Section 3, we develop theory for the privacy cost under a single exact fairness constraint and then generalize this theory to some extent. By simulating a risk-parity linear regression problem in Section 4, we validate our theory and illustrate the utility of our tools. Finally, we summarize our work in Section 5 and point out an interesting avenue of future work.

## 1.1 RELATED WORK

The interaction between fairness and privacy has been investigated from three perspectives: learning approximately fair models without sensitive attributes (Hashimoto et al., 2018; Lahoti et al., 2020), learning approximately fair models with wildly noisy sensitive attributes (Kallus et al., 2019; Awasthi et al., 2020; Wang et al., 2020), and learning exactly fair models with *structured* noisy sensitive attributes (Lamy et al., 2020; Mozannar et al., 2020). This paper focuses on the third aspect.

The works that are most pertinent to ours are Lamy et al. (2020) and Mozannar et al. (2020). Lamy et al. (2020) assume that the sensitive attributes are subject to noise from the mutually contaminated learning model. Under such a structured noise mechanism, the noise rates can be consistently estimated, and when enforcing fairness with regard to noisy groups, scaling the fairness tolerance parameter more tightly is all that is required. Mozannar et al. (2020) suggest a differentially private model to release the sensitive attributes, which is a special type of the mutually contaminated learning model. Under such a designed noise mechanism, Mozannar et al. (2020) show that if the classifier is independent of the sensitive attributes, then exact fairness with regard to noisy sensitive attributes is equivalent to that with regard to true sensitive attributes. The idea of the equivalence can be found in Lamy et al. (2020) while Mozannar et al. (2020) put it into a formal statement.

We basically study the statistical cost of privacy on the generalizability of fairness when using Lamy et al. (2020)'s method under Mozannar et al. (2020)'s privacy mechanism.

## 2 PROBLEM SETUP

### 2.1 FAIR MACHINE LEARNING AS CONSTRAINED STOCHASTIC OPTIMIZATION

In-processing fair machine learning is typically a supervised learning problem with fairness constraints (Zafar et al., 2017; Agarwal et al., 2018). Such a problem can most often be formulated as a constrained stochastic optimization problem: (empirical) risk minimization subject to (empirical) fairness constraints.

Consider a fair binary classification problem. Let $\mathcal{X} \subset \mathbb{R}^d$ be the input space, $\mathcal{Y} = \{0, 1\}$ be the set of possible labels, and $\mathcal{A}$ be the set of possible values of the protected/sensitive attribute. In this setup, training and test examples are tuples of the form $(X, A, Y) \in \mathcal{X} \times \mathcal{A} \times \mathcal{Y}$, and a classifier is a map $f : \mathcal{X} \to \{0, 1\}$. Two popular definitions of algorithmic fairness for binary classification are *demographic parity* (Dwork et al., 2011) and *equality of opportunity* (Hardt et al., 2016).

**Definition 2.1** (Demographic parity). *Let $\widehat{Y} = f(X)$ be the output of the classifier. Demographic parity entails $\mathbb{P}\{\widehat{Y} = 1 \mid A = a\} = \mathbb{P}\{\widehat{Y} = 1 \mid A = a'\}$ for all $a, a' \in \mathcal{A}$.*

Demographic parity, also known as statistical parity, means that the prediction $\widehat{Y} = f(X)$ is statistically independent of the protected attribute $A$.

**Definition 2.2** (Equality of opportunity). *Let $Y = 1$ be the advantaged label that is associated with a positive outcome and $\widehat{Y} = f(X)$ be the output of the classifier. Equality of opportunity entails $\mathbb{P}\{\widehat{Y} = 1 \mid A = a, Y = 1\} = \mathbb{P}\{\widehat{Y} = 1 \mid A = a', Y = 1\}$ for all $a, a' \in \mathcal{A}$.*

Equality of opportunity, also known as true positive rate parity, means that the prediction $\widehat{Y} = f(X)$ conditioned on the advantaged label $Y = 1$ is statistically independent of the protected attribute $A$.

Given a parametric model space $\mathcal{H} = \{f_\theta(\cdot) : \theta \in \Theta\}$ and loss function $\ell : \Theta \times \mathcal{X} \times \mathcal{Y} \to \mathbb{R}_+$ (where $\Theta \subset \mathbb{R}^d$ is a finite-dimensional parameter space), an in-processing fair ML routine is to minimize the (empirical) risk $\mathbb{E}[\ell(\theta; X, Y)]$ while satisfying some fairness constraints. To keep things simple, we assume there are only two demographic groups; *i.e.* $|\mathcal{A}| = 2$. Without loss of generality, we refer to one group as advantaged ($A = 1$) and the other as disadvantaged ($A = 0$).

Consider fair learning with demographic parity as an example. At the population level, the goal is to solve the problem:

$$\theta^\star \in \begin{cases} \arg\min_{\theta \in \Theta} & \mathbb{E}[\ell(\theta; X, Y)] \\ \text{subject to} & \mathbb{E}[\mathbf{1}\{f_\theta(X) = 1\}|A = 1] - \mathbb{E}[\mathbf{1}\{f_\theta(X) = 1\}|A = 0] = 0 \end{cases}, \quad (2.1)$$

where the expectation is with respect to the distribution of tuple $(X, A, Y)$. The true underlying distribution is unknown, so we cannot solve (2.1) directly. Instead, we observe IID training samples $\{(X_i, A_i, Y_i)\}_{i=1}^n$ from the true distribution and solve the empirical version of (2.1):

$$\widehat{\theta}_n \in \begin{cases} \arg\min_{\theta \in \Theta} & \frac{1}{n} \sum_{i=1}^n \ell(\theta; X_i, Y_i) \\ \text{subject to} & \left| \frac{\sum_{i=1}^n \mathbf{1}\{f_\theta(X_i)=1, A_i=1\}}{\sum_{i=1}^n \mathbf{1}\{A_i=1\}} - \frac{\sum_{i=1}^n \mathbf{1}\{f_\theta(X_i)=1, A_i=0\}}{\sum_{i=1}^n \mathbf{1}\{A_i=0\}} \right| \le \alpha_n \end{cases}, \quad (2.2)$$

where $0 < \alpha_n = o(\frac{1}{\sqrt{n}})$ is a slackness term shrinking to zero at a rate faster than $\frac{1}{\sqrt{n}}$. Through the rest of the work, we always let $\alpha_n$ be a positive number of order $o(\frac{1}{\sqrt{n}})$.

## 2.2 Local differential privacy mechanism for releasing sensitive attributes

Consider the randomized response mechanism (Warner, 1965; Kairouz et al., 2014) for releasing privatized sensitive attribute:

$$Q(Z = z \mid A = a) = \begin{cases} \frac{e^\varepsilon}{|\mathcal{A}|-1+e^\varepsilon} & \text{if } z = a \\ \frac{1}{|\mathcal{A}|-1+e^\varepsilon} & \text{if } z \ne a \end{cases} \quad (2.3)$$

for all $a, z \in \mathcal{A}$, where $\varepsilon > 0$ controls the privacy level. The *privatized sensitive attribute* $Z$ of the true sensitive attribute $A$ is defined as $Z = Q(\cdot \mid A)$. In addition, the sampling mechanism $Q$ requires $Z \perp\!\!\!\perp X, Y \mid A$. Then the private dataset $\{(X_i, Z_i, Y_i)\}_{i=1}^n$ is generated from the original dataset $\{(X_i, A_i, Y_i)\}_{i=1}^n$ via the transition kernel $Q$.

The randomized response mechanism (2.3) is a locally $\varepsilon$-differentially private mechanism (Duchi et al., 2013), that is

$$\max_{z,a,a' \in \mathcal{A}} \frac{Q(Z = z \mid A = a)}{Q(Z = z \mid A = a')} \le e^\varepsilon.$$

Here a smaller parameter $\varepsilon$ indicates a stronger privacy guarantee. Moreover, the mechanism $Q$ is considered optimal for distribution estimation under local differential privacy constraints (Kairouz et al., 2014; 2016).

From this point forward (with the exception of the general theory presented in Section 3.1), we assume that there are only two demographic groups, *i.e.* $|\mathcal{A}| = 2$. The mechanism (2.3) becomes

$$Q(Z = z \mid A = a) = \begin{cases} \frac{e^\varepsilon}{1+e^\varepsilon} \triangleq 1 - \gamma & \text{if } z = a \\ \frac{1}{1+e^\varepsilon} \triangleq \gamma & \text{if } z \ne a \end{cases} \quad (2.4)$$

for $a \in \{0, 1\}$, where $\gamma \in [0, 0.5)$. The parameter $\gamma = 0$ (or equivalently $\varepsilon = \infty$) signifies complete lack of privacy, whereas $\gamma \to 0.5$ (or equivalently $\varepsilon \to 0$) corresponds to perfect privacy.

## 2.3 Fair machine learning with private sensitive attributes

The privatized sensitive attribute $Z$ can be served as a noisy proxy for the true sensitive attribute $A$. One may wish to learn a fair classifier by directly enforcing fairness notion on $Z_i$'s, the proxies for $A_i$'s. This approach is feasible and justifiable (at the population level) due to the invariance of exact fairness under local differential privacy.

**Proposition 2.3** (Proposition 1 in Mozannar et al. (2020)). *Consider any exact fairness notion among demographic parity and equality of opportunity. Let $\widehat{Y} = f(X)$ be a binary classifier. Then $\widehat{Y}$ is fair with respect to $A$ if and only if $\widehat{Y}$ is fair with respect to $Z$.*

Proposition 2.3 requires $\widehat{Y}$ is only a function of $X$. Mozannar et al. (2020) shows by construction the existence of a classifier $\widehat{Y} = \widetilde{f}(X, Z)$ which is fair with respect to $Z$ but unfair to $A$.

Now we consider fair ML with private sensitive attributes by (empirical) risk minimization subject to fairness constraints with respect to $Z$. Take fair learning with demographic parity as an example. At the population level, the goal is to solve the problem

$$\theta^\star \in \begin{cases} \arg\min_{\theta \in \Theta} & \mathbb{E}[\ell(\theta; X, Y)] \\ \text{subject to} & \mathbb{E}[\mathbf{1}\{f_\theta(X) = 1\}|Z = 1] - \mathbb{E}[\mathbf{1}\{f_\theta(X) = 1\}|Z = 0] = 0 \end{cases}, \quad (2.5)$$

where the expectation is with respect to the distribution of tuple $(X, Z, Y)$. Here $Z$ is the proxy sensitive attribute but the true sensitive attribute $A$ is unobservable. The true underlying distribution is unknown, so we cannot solve (2.5) directly. Instead, we observe IID (private) training samples $\{(X_i, Z_i, Y_i)\}_{i=1}^n$ from the true distribution and solve the empirical version of (2.5):

$$\widetilde{\theta}_n \in \begin{cases} \arg\min_{\theta \in \Theta} & \frac{1}{n} \sum_{i=1}^n \ell(\theta; X_i, Y_i) \\ \text{subject to} & \left| \frac{\sum_{i=1}^n \mathbf{1}\{f_\theta(X_i)=1, Z_i=1\}}{\sum_{i=1}^n \mathbf{1}\{Z_i=1\}} - \frac{\sum_{i=1}^n \mathbf{1}\{f_\theta(X_i)=1, Z_i=0\}}{\sum_{i=1}^n \mathbf{1}\{Z_i=0\}} \right| \le \alpha_n \end{cases}. \tag{2.6}$$

A direct corollary of Proposition 2.3 is that (2.1) and (2.5) have exactly the same solution $\theta^\star$ (assuming uniqueness of the solution). One can also show that under regularity conditions both $\widehat{\theta}_n$ and $\widetilde{\theta}_n$, the solution to (2.2) and to (2.6), are $\sqrt{n}$-consistent for $\theta^\star$. We wish to compare the estimating quality of $\widehat{\theta}_n$ and $\widetilde{\theta}_n$, and quantify the quality difference in terms of the privacy level parameter $\gamma$ (or $\varepsilon$) and few other problem-specific parameters.

### 2.4 ASYMPTOTIC RELATIVE EFFICIENCY

In statistics, *consistency* and *efficiency* are popular notions to evaluate the performance of estimators.

**Definition 2.4** (Consistency). *An estimator $\widehat{\theta}_n$ is consistent for $\theta^\star$ if $\widehat{\theta}_n \xrightarrow{p} \theta^\star$ as $n \to \infty$.*

Suppose that we have two consistent estimators $\widehat{\theta}_n$ and $\widetilde{\theta}_n$. Both of them are reasonable, but which one should be preferred? To answer this question, we can employ the notion of efficiency, *i.e.* measuring how spread out about $\widehat{\theta}_n$ (or $\widetilde{\theta}_n$) is the sampling distribution of the estimator. In light of this, we now adapt the concept of statistical efficiency to fair machine learning.

In fair ML, the most important metric to evaluate the performance of a classifier is fairness violation. Let $c : \Theta \to \mathbb{R}$ be the constraint function. For example, demographic parity constraint corresponds to $c(\theta) = \mathbb{E}\big[\mathbf{1}\{f_\theta(X) = 1\}|A = 1\big] - \mathbb{E}\big[\mathbf{1}\{f_\theta(X) = 1\}|A = 0\big]$. Since the exact fairness notion entails a classifier $f_\theta$ is fair if $c(\theta) = 0$, we define the *(signed) fairness violation* of $\theta$ as $c(\theta)$ itself.

**Definition 2.5** (Efficiency in terms of constraint violations). *Suppose that we have two consistent estimators $\widehat{\theta}_n$ and $\widetilde{\theta}_n$ satisfying*

$$\sqrt{n}\{c(\widehat{\theta}_n) - c(\theta^\star)\} \xrightarrow{d} \mathcal{N}(0, \sigma^2) \text{ and } \sqrt{n}\{c(\widetilde{\theta}_n) - c(\theta^\star)\} \xrightarrow{d} \mathcal{N}(0, \widetilde{\sigma}^2)$$

*as $n \to \infty$. We say that the estimator $\widehat{\theta}_n$ is more efficient (in terms of constraint violations) than $\widetilde{\theta}_n$ if $\sigma^2 \le \widetilde{\sigma}^2$. The asymptotic relative efficiency (ARE) of $\widetilde{\theta}_n$ to $\widehat{\theta}_n$ is*

$$\mathrm{ARE}(\widetilde{\theta}_n, \widehat{\theta}_n) \triangleq \frac{\sigma^2}{\widetilde{\sigma}^2}.$$

*In other words, the estimator $\widehat{\theta}_n$ is more efficient than $\widetilde{\theta}_n$ if $\mathrm{ARE}(\widetilde{\theta}_n, \widehat{\theta}_n) \le 1$.*

Another way to examine the efficiency loss is to look at the asymptotic joint distribution of $c(\widehat{\theta}_n)$ and $c(\widetilde{\theta}_n)$. Let $\rho$ be the asymptotic correlation between $c(\widehat{\theta}_n)$ and $c(\widetilde{\theta}_n)$. The fairness violations of the two estimators can be compared using the ratio of $c(\widehat{\theta}_n)$ to $c(\widetilde{\theta}_n)$, which converges in distribution to a Cauchy random variable $U$:

$$\frac{c(\widehat{\theta}_n)}{c(\widetilde{\theta}_n)} \xrightarrow{d} U \sim p_U(u) = \frac{1}{\pi} \frac{\beta}{(u - \alpha)^2 + \beta^2} \text{ with } \alpha = \frac{\rho\sigma}{\widetilde{\sigma}}, \beta = \frac{\sigma}{\widetilde{\sigma}}\sqrt{1 - \rho^2}.$$

Constraint violation *inflates* if we observe a value of the ratio $|c(\widehat{\theta}_n)/c(\widetilde{\theta}_n)|$ less than one. Assume $\widetilde{\theta}_n$ is more efficient than $\widehat{\theta}_n$, *i.e.* $\sigma^2 < \widetilde{\sigma}^2$. Since $|\rho| \le 1$, the median and mode of $U$, $\alpha$, satisfies $|\alpha| < 1$, which indicates a high likelihood of constraint violation inflation. Precisely, the asymptotic probability of constraint violation inflation is

$$\lim_{n \to \infty} \mathbb{P}\left(\left|\frac{c(\widehat{\theta}_n)}{c(\widetilde{\theta}_n)}\right| < 1\right) = \frac{1}{\pi}\left\{\tan^{-1}\left(\frac{\widetilde{\sigma} - \rho\sigma}{\sigma\sqrt{1 - \rho^2}}\right) + \tan^{-1}\left(\frac{\widetilde{\sigma} + \rho\sigma}{\sigma\sqrt{1 - \rho^2}}\right)\right\} > \frac{1}{2}.$$

In the rest of the paper, the asymptotic relative efficiency (ARE) is the key quantity of interest, which compares the asymptotic variances of two estimators by $\mathrm{ARE} = \lim_{n \to \infty} \mathrm{Var}[c(\widehat{\theta}_n)] / \mathrm{Var}[c(\widetilde{\theta}_n)]$.

## 3 PRIVACY COST IN FAIR MACHINE LEARNING

In this section, we wish to study $\mathrm{ARE}(\widetilde{\theta}_n, \widehat{\theta}_n)$, the asymptotic relative efficiency (ARE) of $\widetilde{\theta}_n$ to $\widehat{\theta}_n$ given by solving (2.6) and (2.2). To this end, we extend the notion of demographic parity and equality of opportunity to a more general form: we say that $\theta$ is fair (with respect to $A$) if

$$c(\theta) \triangleq \frac{\mathbb{E}\big[g(\theta; X, Y)|A = 1\big]}{\mathbb{E}\big[h(X, Y)|A = 1\big]} - \frac{\mathbb{E}\big[g(\theta; X, Y)|A = 0\big]}{\mathbb{E}\big[h(X, Y)|A = 0\big]} = 0. \tag{3.1}$$

The fairness notion (3.1) is known as *linear-fractional* fairness constraint (Celis et al., 2021). Note that demographic parity is a special case of (3.1) if we take $g(\theta; X, Y) = \mathbf{1}\{f_\theta(X) = 1\}$ and $h \equiv 1$. Besides, (3.1) becomes equality of opportunity if we take $g(\theta; X, Y) = \mathbf{1}\{f_\theta(X) = 1, Y = 1\}$ and $h(X, Y) = \mathbf{1}\{Y = 1\}$. When $h \equiv 1$, (3.1) degenerates to *linear* fairness (see Appendix A).

Let the marginal distribution of $A$ and conditional distribution of $(X, Y)$ given $A$ be

$$\begin{cases} \mathbb{P}(A = 0) = \pi_0, & \mathbb{P}(A = 1) = \pi_1 \\ (X, Y)|A = 0 \sim Q_0, & (X, Y)|A = 1 \sim Q_1 \end{cases}. \tag{3.2}$$

Then the distribution of $(X, A, Y)$ is uniquely identified by (3.2). Moreover, $(X, Y) \sim \pi_0 Q_0 + \pi_1 Q_1$ is a mixture of $Q_0$ and $Q_1$ weighted by $\pi_0$ and $\pi_1$. Denote the marginal distribution of $Z$ and conditional distribution of $(X, Y)$ given $Z$ by $\mathbb{P}(Z = k) = \widetilde{\pi}_k, (X, Y)|Z = k \sim \widetilde{Q}_k$ for $k \in \{0, 1\}$. Enforcing fairness notion (3.1) with respect to $Z$ is

$$\widetilde{c}(\theta) \triangleq \frac{\mathbb{E}\big[g(\theta; X, Y)|Z = 1\big]}{\mathbb{E}\big[h(X, Y)|Z = 1\big]} - \frac{\mathbb{E}\big[g(\theta; X, Y)|Z = 0\big]}{\mathbb{E}\big[h(X, Y)|Z = 0\big]} = 0.$$

By some algebra, we find that the proxy constraint function $\widetilde{c}(\theta)$ is equal to the true constraint function $c(\theta)$ up to a scaling factor: $\widetilde{c}(\theta) = \psi_{\mathrm{frac}}(\gamma, \pi_0, \pi_1, m_0, m_1) \times c(\theta)$, where

$$\psi_{\mathrm{frac}}(\gamma, \pi_0, \pi_1, m_0, m_1) \triangleq \frac{(1 - 2\gamma)\pi_0 \pi_1 m_0 m_1}{\{\gamma \pi_0 m_0 + (1 - \gamma)\pi_1 m_1\}\{(1 - \gamma)\pi_0 m_0 + \gamma \pi_1 m_1\}},$$

as well $m_0 \triangleq \mathbb{E}_{Q_0}\big[h(X, Y)\big]$ and $m_1 \triangleq \mathbb{E}_{Q_1}\big[h(X, Y)\big]$. This also implies $c(\theta) = 0$ if and only if $\widetilde{c}(\theta) = 0$, offering an alternative proof for Proposition 2.3 and extending Proposition 2.3 to linear-fractional fairness notions (3.1).

Now we are ready to show the privacy cost in linear-fractional fairness (3.1)-aware learning. First, let the true parameter $\theta^\star$, *i.e.* the solution to the population problem, be

$$\theta^\star \in \begin{cases} \arg\min_{\theta \in \Theta} & \mathbb{E}\big[\ell(\theta; X, Y)\big] \\ \text{subject to} & \dfrac{\mathbb{E}\big[g(\theta; X, Y)|A = 1\big]}{\mathbb{E}\big[h(X, Y)|A = 1\big]} - \dfrac{\mathbb{E}\big[g(\theta; X, Y)|A = 0\big]}{\mathbb{E}\big[h(X, Y)|A = 0\big]} = 0 \end{cases}, \tag{3.3}$$

where the expectation is with respect to the underlying distribution of tuple $(X, A, Y)$.

Then, let the estimator $\widehat{\theta}_n$ be the solution to the empirical problem given the true sensitive attribute,

$$\widehat{\theta}_n \in \begin{cases} \arg\min_{\theta \in \Theta} & \frac{1}{n}\sum_{i=1}^n \ell(\theta; X_i, Y_i) \\ \text{subject to} & \left|\dfrac{\sum_{i=1}^n g(\theta; X_i, Y_i)\mathbf{1}\{A_i = 1\}}{\sum_{i=1}^n h(X_i, Y_i)\mathbf{1}\{A_i = 1\}} - \dfrac{\sum_{i=1}^n g(\theta; X_i, Y_i)\mathbf{1}\{A_i = 0\}}{\sum_{i=1}^n h(X_i, Y_i)\mathbf{1}\{A_i = 0\}}\right| \leq \alpha_n \end{cases}.$$

Finally, let $\widetilde{\theta}_n$ be the solution to the empirical problem given the proxy sensitive attribute,

$$\widetilde{\theta}_n \in \begin{cases} \arg\min_{\theta \in \Theta} & \frac{1}{n}\sum_{i=1}^n \ell(\theta; X_i, Y_i) \\ \text{subject to} & \left|\dfrac{\sum_{i=1}^n g(\theta; X_i, Y_i)\mathbf{1}\{Z_i = 1\}}{\sum_{i=1}^n h(X_i, Y_i)\mathbf{1}\{Z_i = 1\}} - \dfrac{\sum_{i=1}^n g(\theta; X_i, Y_i)\mathbf{1}\{Z_i = 0\}}{\sum_{i=1}^n h(X_i, Y_i)\mathbf{1}\{Z_i = 0\}}\right| \leq \alpha_n \end{cases}.$$

We made the following technical assumptions on the population problem (3.3).

1. **smoothness and concentration:** $\ell$ and $g$ are twice continuously differentiable with respect to $\theta$, and $\ell(\theta^\star; X, Y)$, $\nabla\ell(\theta^\star; X, Y)$, $g(\theta^\star; X, Y)$, $\nabla g(\theta^\star; X, Y)$, $h(X, Y)$ are sub-Gaussian.
2. **uniqueness:** the stochastic optimization problem with a single expected value constraint (3.3) has unique optimal primal-dual pair $(\theta^\star, \lambda^\star)$, and $\theta^\star$ belongs to the interior of the compact set $\Theta$.
3. **positive definiteness:** The Hessian of the Lagrangian evaluated at $(\theta^\star, \lambda^\star)$ is positive definite.

The preceding assumptions are not the most general, but they are easy to interpret. The smoothness conditions on $\ell$ and $g$ with respect to $\theta$, the concentration conditions of $\ell(\theta^\star)$, $g(\theta^\star)$ and $h$, and the uniqueness condition facilitate the use of standard tools from asymptotic statistics to study the large sample properties of the constraint value. The positive definiteness condition postulates the Lagrangian of the equality constrained optimization problem is locally strongly convex at $(\theta^\star, \lambda^\star)$.

The main technical result characterizes the efficiency of $\widehat{\theta}_n$ and $\widetilde{\theta}_n$ (see proof in Appendix C).

**Theorem 3.1** (Privacy cost in linear-fractional fairness (3.1)-aware learning). *Under the standing assumptions, let estimators $\widehat{\theta}_n$ and $\widetilde{\theta}_n$ be consistent for $\theta^\star$, then*

$$\sqrt{n}\{c(\widehat{\theta}_n) - c(\theta^\star)\} \xrightarrow{d} \mathcal{N}(0, \sigma^2) \text{ and } \sqrt{n}\{c(\widetilde{\theta}_n) - c(\theta^\star)\} \xrightarrow{d} \mathcal{N}(0, \widetilde{\sigma}^2),$$

*where*

$$\sigma^2 = \frac{\mathrm{Var}_{Q_0}[g(\theta^\star; X, Y) - \kappa h(X, Y)]}{\pi_0(\mathbb{E}_{Q_0}[h(X, Y)])^2} + \frac{\mathrm{Var}_{Q_1}[g(\theta^\star; X, Y) - \kappa h(X, Y)]}{\pi_1(\mathbb{E}_{Q_1}[h(X, Y)])^2},$$

$$\widetilde{\sigma}^2 = \psi_{\mathrm{frac}}^{-2} \times \left\{ \frac{\mathrm{Var}_{\widetilde{Q}_0}[g(\theta^\star; X, Y) - \kappa h(X, Y)]}{\widetilde{\pi}_0(\mathbb{E}_{\widetilde{Q}_0}[h(X, Y)])^2} + \frac{\mathrm{Var}_{\widetilde{Q}_1}[g(\theta^\star; X, Y) - \kappa h(X, Y)]}{\widetilde{\pi}_1(\mathbb{E}_{\widetilde{Q}_1}[h(X, Y)])^2} \right\},$$

*and*

$$\kappa \triangleq \frac{\mathbb{E}_{Q_0}[g(\theta^\star; X, Y)]}{\mathbb{E}_{Q_0}[h(X, Y)]} = \frac{\mathbb{E}_{Q_1}[g(\theta^\star; X, Y)]}{\mathbb{E}_{Q_1}[h(X, Y)]}.$$

*The asymptotic relative efficiency (ARE) of $\widetilde{\theta}_n$ to $\widehat{\theta}_n$ is*

$$\mathrm{ARE}(\widetilde{\theta}_n, \widehat{\theta}_n) = \varphi\left(\gamma, \frac{\pi_0 m_0}{\pi_1 m_1}, \frac{\mathrm{Var}[g(\theta^\star; X, Y) - \kappa h(X, Y)|A = 0]/m_0}{\mathrm{Var}[g(\theta^\star; X, Y) - \kappa h(X, Y)|A = 1]/m_1}\right), \tag{3.4}$$

*where*

$$\varphi(\gamma, r_1, r_2) \triangleq \frac{(1 - 2\gamma)^2 r_1(r_1 + r_2)}{\{\gamma r_1 + (1 - \gamma)\}^2\{(1 - \gamma)r_1 r_2 + \gamma\} + \{(1 - \gamma)r_1 + \gamma\}^2\{\gamma r_1 r_2 + (1 - \gamma)\}}.$$

Recall that demographic parity corresponds to $h \equiv 1$ and equality of opportunity corresponds to $h(X, Y) = \mathbf{1}\{Y = 1\}$. In order to interpret (3.4), we therefore take $h(X, Y) = \mathbf{1}\{\mathcal{E}(X, Y)\}$, where $\mathcal{E}(X, Y)$ is an event of $X$ and $Y$. Then the ARE (3.4) becomes

$$\mathrm{ARE}(\widetilde{\theta}_n, \widehat{\theta}_n) = \varphi\left(\gamma, \frac{\mathbb{P}(\mathcal{E}(X, Y), A = 0)}{\mathbb{P}(\mathcal{E}(X, Y), A = 1)}, \frac{\mathrm{Var}[g(\theta^\star; X, Y)|\mathcal{E}(X, Y), A = 0]}{\mathrm{Var}[g(\theta^\star; X, Y)|\mathcal{E}(X, Y), A = 1]}\right).$$

Note that the ARE is jointly determined by the level of privacy, a ratio of marginal probabilities of the minority and majority groups, and a ratio of their conditional variances. Theorem 3.1 demonstrates that the cost of privacy is the efficiency loss in terms of fairness violations. For fixed ratios

$$r_1 \triangleq \frac{\mathbb{P}(\mathcal{E}(X, Y), A = 0)}{\mathbb{P}(\mathcal{E}(X, Y), A = 1)} > 0,$$

and

$$r_2 \triangleq \frac{\mathrm{Var}[g(\theta^\star; X, Y)|\mathcal{E}(X, Y), A = 0]}{\mathrm{Var}[g(\theta^\star; X, Y)|\mathcal{E}(X, Y), A = 1]} > 0,$$

function $\varphi(\gamma, r_1, r_2)$ is decreasing in $\gamma$. In the absence of privacy, $\varphi(0, r_1, r_2) = 1$ means no efficiency loss. Under perfect privacy, $\varphi(0.5, r_1, r_2) = 0$ indicates total loss of efficiency. Moreover, $\widehat{\theta}_n$ is always more efficient than $\widetilde{\theta}_n$ because $\mathrm{ARE}(\widetilde{\theta}_n, \widehat{\theta}_n) \leq 1$.

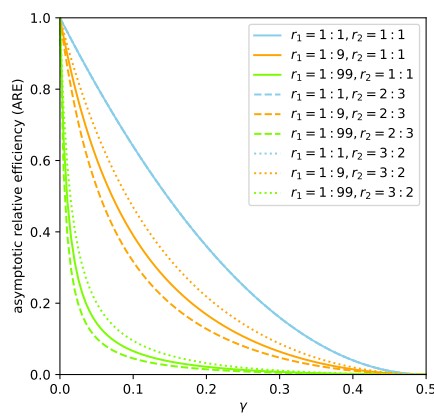

Figure 1: Asymptotic relative efficiency curve of $\gamma$ for varying $r_1$ and $r_2$.

Figure 1 demonstrates the asymptotic relative efficiency (ARE) curve of privacy level $\gamma$ for varying ratios $r_1$ and $r_2$. The ARE is always upper bounded by $(1 - 2\gamma)^2$, which is achieved only if $r_1 = 1$. Therefore for any fixed $\gamma$ and $r_2$, the ARE achieves its maximum only if the dataset is balanced in the sense that $\mathbb{P}(\mathcal{E}(X, Y), A = 0) = \mathbb{P}(\mathcal{E}(X, Y), A = 1)$. Moreover, for any fixed $\gamma$ and $r_2$, the ARE is strictly increasing in $r_1$ (assuming $r_1 \leq 1$). This implies the effect of subpopulation size imbalance: demographic group imbalance degrades the efficiency loss in privately fair learning. In the literature, the effect of group size imbalance on the difficulty of learning fair classifier from contaminated data (note that private sensitive attribute is a particular type of data contamination) was also reported in Konstantinov & Lampert (2022) and the references therein. Lastly, the ARE is strictly increasing in the problem-specific parameter $r_2$, given fixed $\gamma$ and $r_1 < 1$.

### 3.1 GENERAL THEORY

In this subsection, we discuss some extensions to the established theory.

**Multiple demographic groups.** It is natural to extend our theory of two demographic groups to general number of groups. Suppose we have $K + 1(K \geq 2)$ groups indexed by $0, 1, \ldots, K$. The notion of linear-fractional fairness (3.1) can be adapted to more than two groups: we say $\theta$ is fair if

$$\frac{\mathbb{E}\big[g(\theta; X, Y)|A = k\big]}{\mathbb{E}\big[h(X, Y)|A = k\big]} - \frac{\mathbb{E}\big[g(\theta; X, Y)|A = 0\big]}{\mathbb{E}\big[h(X, Y)|A = 0\big]} = 0 \quad \text{for } k \in [K], \tag{3.5}$$

where group $0$ is referred to as a reference group. Let the marginal distribution of $A$ and conditional distribution of $(X, Y)$ given $A$ be

$$\mathbb{P}(A = k) = \pi_k, (X, Y)|A = k \sim Q_k \quad \text{for } k \in \{0\} \cup [K]. \tag{3.6}$$

Then the distribution of $(X, A, Y)$ is uniquely identified by (3.6). Moreover, the distribution of $(X, Y) \sim \sum_{k=0}^{K} \pi_k Q_k \overset{d}{\triangleq} Q_\star$ is a mixture of $Q_k$'s weighted by $\pi_k$'s.

Let the private mechanism $Q$ be

$$Q(Z = z \mid A = a) = \begin{cases} \frac{e^\varepsilon}{K + e^\varepsilon} \triangleq 1 - K\gamma & \text{if } z = a \\ \frac{1}{K + e^\varepsilon} \triangleq \gamma & \text{if } z \neq a \end{cases}$$

where $\gamma \in \left[0, \frac{1}{K+1}\right)$. The mechanism $Q$ perturbs the membership of a group to a different group that is evenly picked at random from the other groups. The parameter $\gamma = 0$ (or equivalently $\varepsilon = \infty$) signifies complete lack of privacy, whereas $\gamma \to \frac{1}{K+1}$ (or equivalently $\varepsilon \to 0$) means perfect privacy.

The joint distribution of $(X, Z, Y)$ is uniquely identified by the marginal distribution and conditional distribution as follows:

$$\begin{cases} \mathbb{P}(Z = k) = \gamma + (1 - |\mathcal{A}|\gamma)\pi_k \triangleq \widetilde{\pi}_k \\ (X, Y)|Z = k \sim \frac{\gamma}{\gamma + (1 - |\mathcal{A}|\gamma)\pi_k} Q_\star + \frac{1 - |\mathcal{A}|\gamma}{\gamma + (1 - |\mathcal{A}|\gamma)\pi_k} Q_k \triangleq \widetilde{Q}_k \end{cases} \quad \text{for } k \in \{0\} \cup [K]. \tag{3.7}$$

Let the true parameter $\theta^\star$, *i.e.* the solution to the population problem, be

$$\theta^\star \in \begin{cases} \arg\min_{\theta \in \Theta} & \mathbb{E}\big[\ell(\theta; X, Y)\big] \\ \text{subject to} & \left\{ \frac{\mathbb{E}\big[g(\theta; X, Y)|A = k\big]}{\mathbb{E}\big[h(X, Y)|A = k\big]} - \frac{\mathbb{E}\big[g(\theta; X, Y)|A = 0\big]}{\mathbb{E}\big[h(X, Y)|A = 0\big]} = 0 \right\}_{k=1}^{K} \end{cases},$$

where the expectation is with respect to the underlying distribution of tuple $(X, A, Y)$.

Then, let the estimator $\widehat{\theta}_n$ be the solution to the empirical problem given the true sensitive attribute,

$$\widehat{\theta}_n \in \begin{cases} \arg\min_{\theta \in \Theta} & \frac{1}{n} \sum_{i=1}^{n} \ell(\theta; X_i, Y_i) \\ \text{subject to} & \left\{ \left| \frac{\sum_{i=1}^{n} g(\theta; X_i, Y_i)\mathbf{1}\{A_i = k\}}{\sum_{i=1}^{n} h(X_i, Y_i)\mathbf{1}\{A_i = k\}} - \frac{\sum_{i=1}^{n} g(\theta; X_i, Y_i)\mathbf{1}\{A_i = 0\}}{\sum_{i=1}^{n} h(X_i, Y_i)\mathbf{1}\{A_i = 0\}} \right| \leq \alpha_n \right\}_{k=1}^{K} \end{cases}.$$

Finally, let $\widetilde{\theta}_n$ be the solution to the empirical problem given the proxy sensitive attribute,

$$\widetilde{\theta}_n \in \begin{cases} \arg\min_{\theta\in\Theta} & \frac{1}{n}\sum_{i=1}^n \ell(\theta; X_i, Y_i) \\ \text{subject to} & \left\{ \left| \frac{\sum_{i=1}^n g(\theta; X_i, Y_i)\mathbf{1}\{Z_i=k\}}{\sum_{i=1}^n h(X_i, Y_i)\mathbf{1}\{Z_i=k\}} - \frac{\sum_{i=1}^n g(\theta; X_i, Y_i)\mathbf{1}\{Z_i=0\}}{\sum_{i=1}^n h(X_i, Y_i)\mathbf{1}\{Z_i=0\}} \right| \le \alpha_n \right\}_{k=1}^K \end{cases}.$$

The true fairness constraint function $\boldsymbol{c}(\theta) : \mathbb{R}^d \to \mathbb{R}^K$ is defined as

$$\boldsymbol{c}(\theta) \triangleq (c_1(\theta), \dots, c_K(\theta))^\top \text{ with } c_k(\theta) = \frac{\mathbb{E}\big[g(\theta; X, Y)|A = k\big]}{\mathbb{E}\big[h(X, Y)|A = k\big]} - \frac{\mathbb{E}\big[g(\theta; X, Y)|A = 0\big]}{\mathbb{E}\big[h(X, Y)|A = 0\big]}, k \in [K].$$

Under the same assumptions as the two-group problem, we have the main technical result as follows (see Appendix D for a complete treatment to the general-number-of-groups problem).

**Theorem 3.2** (Privacy cost in linear-fractional fairness (3.5)-aware learning). *Under the standing assumptions, let estimators $\widehat{\theta}_n$ and $\widetilde{\theta}_n$ be consistent for $\theta^\star$, then*

$$\sqrt{n}\{\boldsymbol{c}(\widehat{\theta}_n) - \boldsymbol{c}(\theta^\star)\} \xrightarrow{d} \mathcal{N}(\mathbf{0}, \Sigma) \text{ and } \sqrt{n}\{\boldsymbol{c}(\widetilde{\theta}_n) - \boldsymbol{c}(\theta^\star)\} \xrightarrow{d} \mathcal{N}(\mathbf{0}, \Psi_{\mathrm{frac}}^{-1}\widetilde{\Sigma}\Psi_{\mathrm{frac}}^{-\top}),$$

*where*

$$\Sigma_{kl} = \frac{\mathrm{Var}_{Q_0}[g(\theta^\star; X, Y) - \kappa h(X, Y)]}{\pi_0(\mathbb{E}_{Q_0}[h(X, Y)])^2} + \left( \frac{\mathrm{Var}_{Q_k}[g(\theta^\star; X, Y) - \kappa h(X, Y)]}{\pi_k(\mathbb{E}_{Q_k}[h(X, Y)])^2} \right) \mathbf{1}\{k = l\}$$

$$\widetilde{\Sigma}_{kl} = \frac{\mathrm{Var}_{\widetilde{Q}_0}[g(\theta^\star; X, Y) - \kappa h(X, Y)]}{\widetilde{\pi}_0(\mathbb{E}_{\widetilde{Q}_0}[h(X, Y)])^2} + \left( \frac{\mathrm{Var}_{\widetilde{Q}_k}[g(\theta^\star; X, Y) - \kappa h(X, Y)]}{\widetilde{\pi}_k(\mathbb{E}_{\widetilde{Q}_k}[h(X, Y)])^2} \right) \mathbf{1}\{k = l\}$$

*for $k, l \in [K]$, and*

$$\kappa \triangleq \frac{\mathbb{E}_{Q_0}[g(\theta^\star; X, Y)]}{\mathbb{E}_{Q_0}[h(X, Y)]} = \frac{\mathbb{E}_{Q_1}[g(\theta^\star; X, Y)]}{\mathbb{E}_{Q_1}[h(X, Y)]} = \cdots = \frac{\mathbb{E}_{Q_K}[g(\theta^\star; X, Y)]}{\mathbb{E}_{Q_K}[h(X, Y)]}.$$

**Missing sensitive attributes.** Some users may choose not to disclose their demographic identities during data collection due to privacy concerns. We investigate how the absence of sensitive attributes impacts the generalizability of fairness constraints. Consider the following missing data mechanism for sensitive attributes :

$$\mathbb{P}(R = 1 \mid X, A, Y) = \mathbb{P}(R = 1 \mid A) \triangleq \omega_A. \tag{3.8}$$

where $R = 1$ corresponds to response (*i.e.*, $A$ is observed) and otherwise $R = 0$ corresponds to non-response (*i.e.*, $A$ is missing). The missingness mechanism (3.8) is a particular type of missing at random (MAR) at the population level and missing completely at random (MCAR) within each subpopulation. One common approach for analyzing data with missing values is to just use the completely observed samples (i.e., samples with all features observed) and discard the samples with some missing features. We employ this strategy by solving the following empirical problem:

$$\widetilde{\theta}_n \in \begin{cases} \arg\min_{\theta\in\Theta} & \frac{1}{n}\sum_{i=1}^n \ell(\theta; X_i, Y_i) \\ \text{subject to} & \left| \frac{\sum_{i=1}^n g(\theta; X_i, Y_i)\mathbf{1}\{A_i=1, R_i=1\}}{\sum_{i=1}^n h(X_i, Y_i)\mathbf{1}\{A_i=1, R_i=1\}} - \frac{\sum_{i=1}^n g(\theta; X_i, Y_i)\mathbf{1}\{A_i=0, R_i=1\}}{\sum_{i=1}^n h(X_i, Y_i)\mathbf{1}\{A_i=0, R_i=1\}} \right| \le \alpha_n \end{cases},$$

of which the empirical risk function is computed with all samples while the fairness constraint function is calculated with samples that include the sensitive attribute. With the same assumptions as the two-group problem and further assuming that the response probability is non-vanishing, *i.e.*, $\omega_a > 0$ for $a \in \{0, 1\}$, we have the asymptotic relative efficiency (ARE) of $\widetilde{\theta}_n$ to $\widehat{\theta}_n$ as follows (see Appendix E for a complete treatment to the missing sensitive attributes problem):

$$\mathrm{ARE}(\widetilde{\theta}_n, \widehat{\theta}_n) = \frac{r_2 + r_1}{\omega_0^{-1} r_2 + \omega_1^{-1} r_1}, r_1 = \frac{\pi_0 m_0}{\pi_1 m_1}, r_2 = \frac{\mathrm{Var}[g(\theta^\star; X, Y) - \kappa h(X, Y)|A = 0]/m_0}{\mathrm{Var}[g(\theta^\star; X, Y) - \kappa h(X, Y)|A = 1]/m_1},$$

This indicates that any probability of missing data degrades the asymptotic efficiency of the estimator inversely proportionally.

## 4 SIMULATIONS

We simulate the asymptotic relative efficiency (ARE) for the risk-parity linear regression problem:

$$\begin{aligned}
&\min_{\beta \in \Theta} && \mathbb{E}\big[(Y - \beta^\top X)^2\big] \\
&\text{subject to} && \mathbb{E}\big[(Y - \beta^\top X)^2|A = 1\big] - \mathbb{E}\big[(Y - \beta^\top X)^2|A = 0\big] = 0
\end{aligned} \tag{4.1}$$

where we generate $n \in \{300, 3000\}$ samples by the following data generating process:

$$A \sim \text{Bernoulli}(1 - \pi_0), X|A = a \sim \mathcal{N}(\mu_a, \Sigma_a) \text{ and } Y|X, A = a \sim \mathcal{N}(\beta_a^\top X, \sigma_a^2)$$

for $a \in \{0, 1\}$. We pick $\mu_0 = (1, 2)^\top, \mu_1 = (2, 1)^\top, \Sigma_0 = \Sigma_1 = I_2, \sigma_0^2 = \sigma_1^2 = 1$ and investigate two scenarios: imbalanced subgroups with $\pi_0 = 0.3$ and balanced subgroups with $\pi_0 = 0.5$. The goal of the optimization problem (4.1) is to minimize the population risk (in least square) while satisfying the parity of subpopulation risks (in least square) of group $A = 0$ and group $A = 1$.

In Figure 2, we plot relative efficiency curves for $\pi_0 = 0.3$ and $\pi_0 = 0.5$, all of which are averaged over 500 replicates. For large sample size $n$, the relative efficiency curves are close to the theoretical line of asymptotic relative efficiency curve, validating our theory in the large sample regime. As a by-product, our theory can visualize the fairness-privacy trade-off without retraining models with varying privacy budgets.

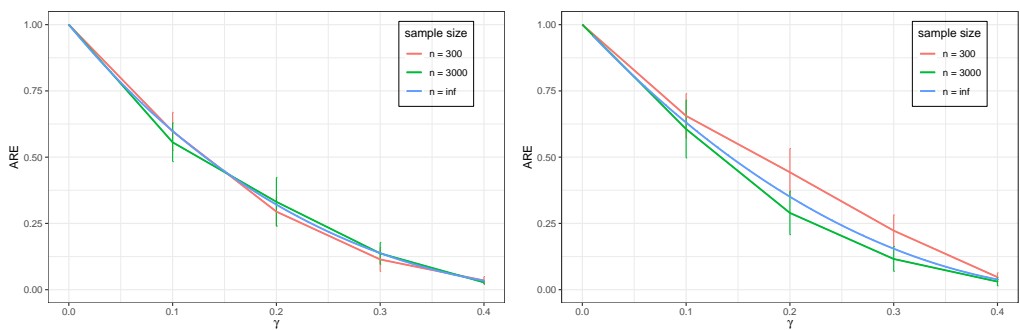

Figure 2: Relative efficiency curves for $\pi_0 = 0.3$ (left) and $\pi_0 = 0.5$ (right).

## 5 SUMMARY AND DISCUSSION

In this work, we study the statistical impact of privacy on fairness under the task of learning fair machine learning models with private sensitive attributes. We define a restricted notion of asymptotic statistical efficiency in order to examine such impact. Quantitatively, the cost of privacy on fairness generalizability is represented by a relative decline in statistical efficiency. The relative efficiency loss is interpretable: it explicitly depends on the privacy budget, subpopulation imbalance level, and a number of other problem-specific quantities. We validate and demonstrate the utility of our theory by a synthetic task of risk-parity linear regression with private group membership.

For the sake of clarity, we consider $h \equiv 1$. Denote the loss vectors with regard to the true sensitive attribute $A$ and the noisy sensitive attribute $Z$, and the Markov transition matrix induced by the privacy mechanism $Q$ (2.4) by

$$L_A(\theta) = \begin{bmatrix} \mathbb{E}\big[g(\theta; X, Y)|A = 1\big] \\ \mathbb{E}\big[g(\theta; X, Y)|A = 0\big] \end{bmatrix}, L_Z(\theta) = \begin{bmatrix} \mathbb{E}\big[g(\theta; X, Y)|Z = 1\big] \\ \mathbb{E}\big[g(\theta; X, Y)|Z = 0\big] \end{bmatrix} \text{ and } M = \begin{bmatrix} 1 - \gamma & \gamma \\ \gamma & 1 - \gamma \end{bmatrix}.$$

Further, let $\boldsymbol{b} = (1, -1)^\top$. Noiseless, noisy, and debiased constraints are equivalent to each other at the population level in the way that $\boldsymbol{b}^\top L_A(\theta) = 0 \iff \boldsymbol{b}^\top L_Z(\theta) = 0 \iff \boldsymbol{b}^\top M^{-1} L_Z(\theta) = 0$. Consider their empirical counterparts, we note that $\boldsymbol{b}^\top \widehat{L}_{Z,n}(\theta) = 0 \iff \boldsymbol{b}^\top M^{-1} \widehat{L}_{Z,n}(\theta) = 0$. Combined with our theory, this empirical level equivalence of two constraints implies that using the inverse of the empirical transition matrix to match the noisy constraint to the noiseless constraint cannot improve the efficiency of the in-processing training procedure. Developing a principled in-processing method to increase the statistical efficiency is an intriguing direction for future research.

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

## A  LINEAR FAIRNESS CONSTRAINT

We extend the notion of demographic parity to a more general form: we say that $\theta$ is fair (with respect to $A$) if

$$\mathbb{E}\big[g(\theta; X, Y)|A = 1\big] - \mathbb{E}\big[g(\theta; X, Y)|A = 0\big] = 0. \tag{A.1}$$

The fairness notion (A.1) is known as *linear* fairness constraint Celis et al. (2021). Note that demographic parity is a special case of (A.1) if we take $g(\theta; X, Y) = \mathbf{1}\{f_\theta(X) = 1\}$.

On the one hand, enforcing fairness notion (A.1) with respect to $A$ is

$$\mathbb{E}_{(X,Y)|A=1}\big[g(\theta; X, Y)\big] - \mathbb{E}_{(X,Y)|A=0}\big[g(\theta; X, Y)\big] = 0$$

or equivalently

$$\mathbb{E}_{Q_1}\big[g(\theta; X, Y)\big] - \mathbb{E}_{Q_0}\big[g(\theta; X, Y)\big] = 0.$$

On the other hand, enforcing fairness notion (A.1) with respect to $Z$ is

$$\mathbb{E}_{(X,Y)|Z=1}\big[g(\theta; X, Y)\big] - \mathbb{E}_{(X,Y)|Z=0}\big[g(\theta; X, Y)\big] = 0$$

or equivalently

$$\mathbb{E}_{\frac{\gamma\pi_0}{\gamma\pi_0+(1-\gamma)\pi_1}Q_0+\frac{(1-\gamma)\pi_1}{\gamma\pi_0+(1-\gamma)\pi_1}Q_1}\big[g(\theta; X, Y)\big] - \mathbb{E}_{\frac{(1-\gamma)\pi_0}{(1-\gamma)\pi_0+\gamma\pi_1}Q_0+\frac{\gamma\pi_1}{(1-\gamma)\pi_0+\gamma\pi_1}Q_1}\big[g(\theta; X, Y)\big] = 0.$$

Therefore, the true fairness constraint function is

$$c(\theta) = \int_{\mathcal{X}\times\mathcal{Y}} g(\theta; x, y)d(Q_1 - Q_0)(x, y),$$

while the proxy fairness constraint function is

$$\begin{aligned}
\widetilde{c}(\theta) &= \left(-\frac{\gamma\pi_0}{\gamma\pi_0 + (1-\gamma)\pi_1} + \frac{(1-\gamma)\pi_0}{(1-\gamma)\pi_0 + \gamma\pi_1}\right)\int_{\mathcal{X}\times\mathcal{Y}} g(\theta; x, y)d(Q_1 - Q_0)(x, y) \\
&= \left(\frac{(1-\gamma)\pi_1}{\gamma\pi_0 + (1-\gamma)\pi_1} - \frac{\gamma\pi_1}{(1-\gamma)\pi_0 + \gamma\pi_1}\right)\int_{\mathcal{X}\times\mathcal{Y}} g(\theta; x, y)d(Q_1 - Q_0)(x, y) \\
&\triangleq \psi_{\text{lin}}(\gamma, \pi_0, \pi_1) \times c(\theta).
\end{aligned} \tag{A.2}$$

By (A.2), the proxy constraint function $\widetilde{c}(\theta)$ is equal to the true $c(\theta)$ up to a scaling factor

$$\begin{aligned}
\psi_{\text{lin}}(\gamma, \pi_0, \pi_1) &\triangleq -\frac{\gamma\pi_0}{\gamma\pi_0 + (1-\gamma)\pi_1} + \frac{(1-\gamma)\pi_0}{(1-\gamma)\pi_0 + \gamma\pi_1} \\
&= \frac{(1-\gamma)\pi_1}{\gamma\pi_0 + (1-\gamma)\pi_1} - \frac{\gamma\pi_1}{(1-\gamma)\pi_0 + \gamma\pi_1} \\
&= \frac{(1-2\gamma)\pi_0\pi_1}{\{\gamma\pi_0 + (1-\gamma)\pi_1\}\{(1-\gamma)\pi_0 + \gamma\pi_1\}}.
\end{aligned} \tag{A.3}$$

This also implies $c(\theta) = 0$ if and only if $\widetilde{c}(\theta) = 0$, providing an alternative proof for Proposition 2.3.

Now we are ready to show the privacy cost in linear fairness (A.1)-aware learning. First, let the true parameter $\theta^\star$, *i.e.* the solution to the population problem, be

$$\theta^\star \in \begin{cases} \arg\min_{\theta\in\Theta} & \mathbb{E}\big[\ell(\theta; X, Y)\big] \\ \text{subject to} & \mathbb{E}\big[g(\theta; X, Y)|A = 1\big] - \mathbb{E}\big[g(\theta; X, Y)|A = 0\big] = 0 \end{cases}, \tag{A.4}$$

where the expectation is with respect to the underlying distribution of tuple $(X, A, Y)$.

Then, let the estimator $\widehat{\theta}_n$ be the solution to the empirical problem given the true sensitive attribute,

$$\widehat{\theta}_n \in \begin{cases} \arg\min_{\theta \in \Theta} & \frac{1}{n} \sum_{i=1}^n \ell(\theta; X_i, Y_i) \\ \text{subject to} & \left| \frac{\sum_{i=1}^n g(\theta; X_i, Y_i) \mathbf{1}\{A_i=1\}}{\sum_{i=1}^n \mathbf{1}\{A_i=1\}} - \frac{\sum_{i=1}^n g(\theta; X_i, Y_i) \mathbf{1}\{A_i=0\}}{\sum_{i=1}^n \mathbf{1}\{A_i=0\}} \right| \le \alpha_n \end{cases}.$$

Finally, let the estimator $\widetilde{\theta}_n$ be the solution to the empirical problem given the proxy sensitive attribute,

$$\widetilde{\theta}_n \in \begin{cases} \arg\min_{\theta \in \Theta} & \frac{1}{n} \sum_{i=1}^n \ell(\theta; X_i, Y_i) \\ \text{subject to} & \left| \frac{\sum_{i=1}^n g(\theta; X_i, Y_i) \mathbf{1}\{Z_i=1\}}{\sum_{i=1}^n \mathbf{1}\{Z_i=1\}} - \frac{\sum_{i=1}^n g(\theta; X_i, Y_i) \mathbf{1}\{Z_i=0\}}{\sum_{i=1}^n \mathbf{1}\{Z_i=0\}} \right| \le \alpha_n \end{cases}.$$

We made the following technical assumptions on the problem (A.4).

1. **smoothness and concentration:** $\ell$ and $g$ are twice continuously differentiable with respect to $\theta$, and $\ell(\theta^\star; X, Y), \nabla\ell(\theta^\star; X, Y), g(\theta^\star; X, Y), \nabla g(\theta^\star; X, Y)$ are sub-Gaussian random variables.
2. **uniqueness:** the stochastic optimization problem with a single expected value constraint (A.4) has a unique optimal primal-dual pair $(\theta^\star, \lambda^\star)$, and $\theta^\star$ belongs to the interior of the compact set $\Theta$.
3. **positive definiteness:** The Hessian of the Lagrangian evaluated at $(\theta^\star, \lambda^\star)$ is positive definite.

We have the main technical result as follows.

**Theorem A.1** (Privacy cost in linear fairness (A.1)-aware learning). *Under the standing assumptions, let estimators $\widehat{\theta}_n$ and $\widetilde{\theta}_n$ be consistent for $\theta^\star$, then*

$$\sqrt{n}\{c(\widehat{\theta}_n) - c(\theta^\star)\} \xrightarrow{d} \mathcal{N}(0, \sigma^2) \text{ and } \sqrt{n}\{c(\widetilde{\theta}_n) - c(\theta^\star)\} \xrightarrow{d} \mathcal{N}(0, \widetilde{\sigma}^2),$$

*where*

$$\sigma^2 = \frac{\text{Var}_{Q_0}[g(\theta^\star; X, Y)]}{\pi_0} + \frac{\text{Var}_{Q_1}[g(\theta^\star; X, Y)]}{\pi_1}$$

*and*

$$\widetilde{\sigma}^2 = \psi_{\text{lin}}^{-2} \times \left\{ \frac{\text{Var}_{\widetilde{Q}_0}[g(\theta^\star; X, Y)]}{\widetilde{\pi}_0} + \frac{\text{Var}_{\widetilde{Q}_1}[g(\theta^\star; X, Y)]}{\widetilde{\pi}_1} \right\}.$$

*The asymptotic relative efficiency (ARE) of $\widetilde{\theta}_n$ to $\widehat{\theta}_n$ is*

$$\text{ARE}(\widetilde{\theta}_n, \widehat{\theta}_n) = \varphi\left( \gamma, \frac{\pi_0}{\pi_1}, \frac{\text{Var}[g(\theta^\star; X, Y)|A=0]}{\text{Var}[g(\theta^\star; X, Y)|A=1]} \right),$$

*where*

$$\varphi(\gamma, r_1, r_2) \triangleq \frac{(1-2\gamma)^2 r_1 (r_1 + r_2)}{\{\gamma r_1 + (1-\gamma)\}^2 \{(1-\gamma)r_1 r_2 + \gamma\} + \{(1-\gamma)r_1 + \gamma\}^2 \{\gamma r_1 r_2 + (1-\gamma)\}}.$$

*Proof of Theorem A.1.* Note that Theorem 3.1 implies Theorem A.1 by letting $h(X, Y) \equiv 1$. Therefore, it is sufficient to prove Theorem 3.1, whose proof can be found in Appendix C. $\square$

Theorem A.1 demonstrates that the cost of privacy is the efficiency loss in terms of fairness violations. For fixed ratios $r_1 \triangleq \pi_0/\pi_1 > 0$ and $r_2 \triangleq \text{Var}[g(\theta^\star; X, Y)|A=0]/\text{Var}[g(\theta^\star; X, Y)|A=1] > 0$, $\varphi_{\text{lin}}(\gamma, r_1, r_2)$ is a decreasing function in $\gamma$. In the absence of privacy, $\varphi_{\text{lin}}(0, r_1, r_2) = 1$ means no efficiency loss. Under perfect privacy, $\varphi_{\text{lin}}(0.5, r_1, r_2) = 0$ indicates total loss of efficiency. Moreover, $\widehat{\theta}_n$ is always more efficient than $\widetilde{\theta}_n$ because $\text{ARE}(\widetilde{\theta}_n, \widehat{\theta}_n) \le 1$.

Figure 3 demonstrates the asymptotic relative efficiency (ARE) curve of privacy level $\gamma$ for varying ratios $r_1$ and $r_2$. The ARE is always upper bounded by $(1-2\gamma)^2$, which is achieved only if $\pi_0 = \pi_1 = 0.5$. Recall that $\pi_0 = \mathbb{P}(A=0)$ and $\pi_1 = \mathbb{P}(A=1)$. Therefore for any fixed $\gamma$ and $r_2$, the ARE achieves its maximum only if the dataset is balanced in the sensitive attribute $A$. Moreover, for any fixed $\gamma$ and $r_2$, the ARE is strictly increasing in $\pi_0$ (assuming $\pi_0 < 0.5$). This implies the effect of subgroup size imbalance: demographic group imbalance degrades the efficiency loss in privately fair learning. Lastly, the ARE is strictly increasing in the problem-specific parameter $r_2$, given fixed $\gamma$ and $r_1 < 1$.

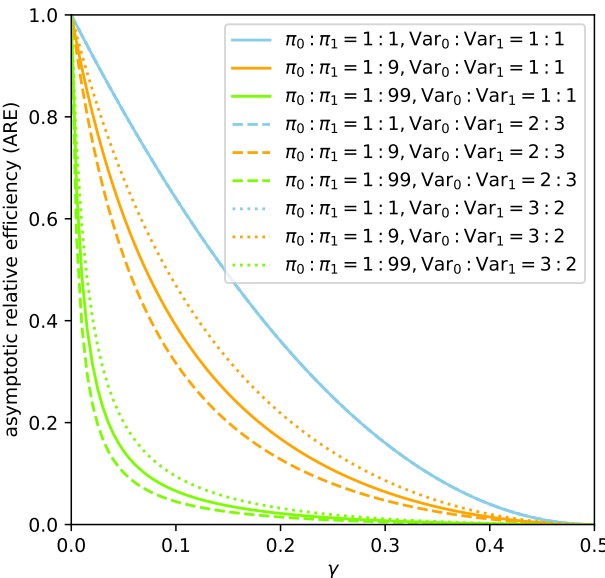

Figure 3: Asymptotic relative efficiency curve of $\gamma$ for varying ratios of $\pi_0$ to $\pi_1$ and $\text{Var}[g(\theta^\star; X, Y)|A = 0]$ to $\text{Var}[g(\theta^\star; X, Y)|A = 1]$.

## B    LINEAR-FRACTIONAL FAIRNESS CONSTRAINT

We provide further discussion to supplement Section 3. Recall the marginal distributions and conditional distributions in (3.2) and

$$\begin{cases} \mathbb{P}(Z = 0) = \widetilde{\pi}_0, & \mathbb{P}(Z = 1) = \widetilde{\pi}_1 \\ (X, Y)|Z = 0 \sim \widetilde{Q}_0, & (X, Y)|Z = 1 \sim \widetilde{Q}_1 \end{cases}.$$

Under the private mechanism $Q$ in (2.4), we have

$$\begin{cases} \widetilde{\pi}_0 = (1 - \gamma)\pi_0 + \gamma\pi_1, \widetilde{\pi}_1 = \gamma\pi_0 + (1 - \gamma)\pi_1 \\ \widetilde{Q}_0 \overset{d}{=} \frac{(1-\gamma)\pi_0}{(1-\gamma)\pi_0 + \gamma\pi_1}Q_0 + \frac{\gamma\pi_1}{(1-\gamma)\pi_0 + \gamma\pi_1}Q_1 \\ \widetilde{Q}_1 \overset{d}{=} \frac{\gamma\pi_0}{\gamma\pi_0 + (1-\gamma)\pi_1}Q_0 + \frac{(1-\gamma)\pi_1}{\gamma\pi_0 + (1-\gamma)\pi_1}Q_1 \end{cases}. \tag{B.1}$$

The marginal distribution and conditional distribution in (B.1) uniquely identify the joint distribution of $(X, Z, Y)$.

On the one hand, enforcing fairness notion (3.1) with respect to $A$ is

$$\frac{\mathbb{E}_{(X,Y)|A=1}\big[g(\theta; X, Y)\big]}{\mathbb{E}_{(X,Y)|A=1}\big[h(X, Y)\big]} - \frac{\mathbb{E}_{(X,Y)|A=0}\big[g(\theta; X, Y)\big]}{\mathbb{E}_{(X,Y)|A=0}\big[h(X, Y)\big]} = 0$$

or equivalently

$$c(\theta) \triangleq \frac{\mathbb{E}_{Q_1}\big[g(\theta; X, Y)\big]}{\mathbb{E}_{Q_1}\big[h(X, Y)\big]} - \frac{\mathbb{E}_{Q_0}\big[g(\theta; X, Y)\big]}{\mathbb{E}_{Q_0}\big[h(X, Y)\big]} = 0.$$

On the other hand, enforcing fairness notion (3.1) with respect to $Z$ is

$$\frac{\mathbb{E}_{(X,Y)|Z=1}\big[g(\theta; X, Y)\big]}{\mathbb{E}_{(X,Y)|Z=1}\big[h(X, Y)\big]} - \frac{\mathbb{E}_{(X,Y)|Z=0}\big[g(\theta; X, Y)\big]}{\mathbb{E}_{(X,Y)|Z=0}\big[h(X, Y)\big]} = 0$$

or equivalently

$$\widetilde{c}(\theta) \triangleq \left\{ \begin{aligned} &\frac{\gamma\pi_0\mathbb{E}_{Q_0}\big[g(\theta; X, Y)\big] + (1 - \gamma)\pi_1\mathbb{E}_{Q_1}\big[g(\theta; X, Y)\big]}{\gamma\pi_0\mathbb{E}_{Q_0}\big[h(X, Y)\big] + (1 - \gamma)\pi_1\mathbb{E}_{Q_1}\big[h(X, Y)\big]} \\ &- \frac{(1 - \gamma)\pi_0\mathbb{E}_{Q_0}\big[g(\theta; X, Y)\big] + \gamma\pi_1\mathbb{E}_{Q_1}\big[g(\theta; X, Y)\big]}{(1 - \gamma)\pi_0\mathbb{E}_{Q_0}\big[h(X, Y)\big] + \gamma\pi_1\mathbb{E}_{Q_1}\big[h(X, Y)\big]} \end{aligned} \right\} = 0.$$

By some algebra, we find that the proxy constraint function $\widetilde{c}(\theta)$ is equal to the true constraint function $c(\theta)$ up to a scaling factor: $\widetilde{c}(\theta) = \psi_{\text{frac}}(\gamma, \pi_0, \pi_1, m_0, m_1) \times c(\theta)$, where

$$\psi_{\text{frac}}(\gamma, \pi_0, \pi_1, m_0, m_1) \triangleq \frac{(1 - 2\gamma)\pi_0\pi_1 m_0 m_1}{\{\gamma\pi_0 m_0 + (1-\gamma)\pi_1 m_1\}\{(1-\gamma)\pi_0 m_0 + \gamma\pi_1 m_1\}}, \quad \text{(B.2)}$$

as well $m_0 \triangleq \mathbb{E}_{Q_0}[h(X, Y)]$ and $m_1 \triangleq \mathbb{E}_{Q_1}[h(X, Y)]$.

By comparing the scaling factor (B.2) with the functional form of (A.3), we can rewrite $\psi_{\text{frac}}(\cdot)$ by

$$\psi_{\text{frac}}(\gamma, \pi_0, \pi_1, m_0, m_1) = \psi_{\text{lin}}(\gamma, \pi_0 m_0, \pi_1 m_1).$$

Therefore, we can interpret the scaling factor $\psi_{\text{frac}}(\cdot)$ by treating $\pi_0 m_0$ and $\pi_1 m_1$ as a whole, allowing us to understand the privacy cost from a different perspective. Note that for equality of opportunity, we have $\pi_a m_a = \mathbb{P}(A = a)\mathbb{P}(Y = 1 | A = a) = \mathbb{P}(Y = 1, A = a)$ for $a \in \{0, 1\}$.

For equality of opportunity, Mozannar et al. (2020) show a sample complexity bound for the fairness violation of the estimator $\widetilde{\theta}_n$:

$$c(\widetilde{\theta}_n) - c(\theta^\star) \le \frac{C_1(1-\gamma)}{(1-2\gamma)p^2}\left(C_2 + C_3\mathcal{R}_{\frac{np}{4}}(\mathcal{F}) + \frac{C_4}{\sqrt{n\delta p}}\right) \quad \text{(B.3)}$$

with probability at least $1 - \delta$, where $p = \min\{\mathbb{P}(Y = 1, A = 0), \mathbb{P}(Y = 1, A = 1)\}$, $\mathcal{R}_{\cdot}(\cdot)$ is the Rademacher complexity, and $C_i$'s $(1 \le i \le 4)$ are some universal constants. Not precisely, the upper bound (B.3) reflects the effect of privacy level via $\gamma$ and the effect of dataset imbalance through $p$. Comparing to this, our theory states that

$$\lim_{n \to \infty} \frac{\text{Var}[c(\widehat{\theta}_n) - c(\theta^\star)]}{\text{Var}[c(\widetilde{\theta}_n) - c(\theta^\star)]} = \varphi\left(\gamma, \frac{\mathbb{P}(Y = 1, A = 0)}{\mathbb{P}(Y = 1, A = 1)}, 1\right),$$

which is depicted by Figure 4.

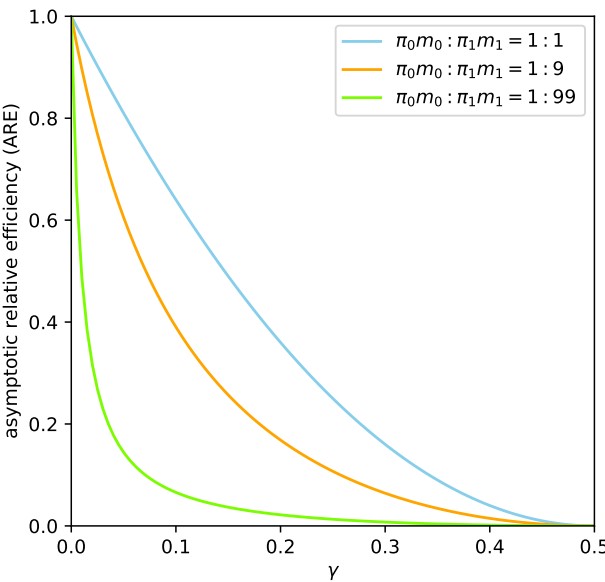

Figure 4: Asymptotic relative efficiency curve of $\gamma$ for varying ratio of $\mathbb{P}(Y = 1, A = 0)$ to $\mathbb{P}(Y = 1, A = 1)$.

## C  PROOF OF THEOREM 3.1

First, we prove the case when $\alpha_n = 0$ for all $n$. For this case both the population problem and the empirical problem are subject to equality constraints.

Consider a stochastic optimization problem with linear-fractional constraint

$$(\mathcal{P}_0): \quad \theta^\star \in \left\{ \begin{array}{ll} \arg\min_{\theta \in \Theta} & \mathbb{E}\big[\ell(\theta; X, Y)\big] \\ \text{subject to} & \dfrac{\mathbb{E}\big[g(\theta; X, Y)|A = 1\big]}{\mathbb{E}\big[h(X, Y)|A = 1\big]} - \dfrac{\mathbb{E}\big[g(\theta; X, Y)|A = 0\big]}{\mathbb{E}\big[h(X, Y)|A = 0\big]} = 0 \end{array} \right\},$$

where the expectation is with respect to the underlying distribution of tuple $(X, A, Y)$.

The corresponding empirical problem given the true sensitive attribute is

$$(\mathcal{P}_n): \quad \widehat{\theta}_n \in \left\{ \begin{array}{ll} \arg\min_{\theta \in \Theta} & \dfrac{1}{n}\sum_{i=1}^{n} \ell(\theta; X_i, Y_i) \\ \text{subject to} & \dfrac{\sum_{i=1}^{n} g(\theta; X_i, Y_i)\mathbf{1}\{A_i = 1\}}{\sum_{i=1}^{n} h(X_i, Y_i)\mathbf{1}\{A_i = 1\}} - \dfrac{\sum_{i=1}^{n} g(\theta; X_i, Y_i)\mathbf{1}\{A_i = 0\}}{\sum_{i=1}^{n} h(X_i, Y_i)\mathbf{1}\{A_i = 0\}} = 0 \end{array} \right\}.$$

The corresponding empirical problem given the proxy sensitive attribute is

$$(\widetilde{\mathcal{P}}_n): \quad \widetilde{\theta}_n \in \left\{ \begin{array}{ll} \arg\min_{\theta \in \Theta} & \dfrac{1}{n}\sum_{i=1}^{n} \ell(\theta; X_i, Y_i) \\ \text{subject to} & \dfrac{\sum_{i=1}^{n} g(\theta; X_i, Y_i)\mathbf{1}\{Z_i = 1\}}{\sum_{i=1}^{n} h(X_i, Y_i)\mathbf{1}\{Z_i = 1\}} - \dfrac{\sum_{i=1}^{n} g(\theta; X_i, Y_i)\mathbf{1}\{Z_i = 0\}}{\sum_{i=1}^{n} h(X_i, Y_i)\mathbf{1}\{Z_i = 0\}} = 0 \end{array} \right\}.$$

We denote

$$F(\theta) = \mathbb{E}\big[\ell(\theta; X, Y)\big], \widehat{F}_n(\theta) = \frac{1}{n}\sum_{i=1}^{n} \ell(\theta; X_i, Y_i), G(\theta) = \frac{\mathbb{E}\big[g(\theta; X, Y)|A = 1\big]}{\mathbb{E}\big[h(X, Y)|A = 1\big]} - \frac{\mathbb{E}\big[g(\theta; X, Y)|A = 0\big]}{\mathbb{E}\big[h(X, Y)|A = 0\big]}$$

and

$$\widehat{G}_n(\theta) = \frac{\sum_{i=1}^{n} g(\theta; X_i, Y_i)\mathbf{1}\{A_i = 1\}}{\sum_{i=1}^{n} h(X_i, Y_i)\mathbf{1}\{A_i = 1\}} - \frac{\sum_{i=1}^{n} g(\theta; X_i, Y_i)\mathbf{1}\{A_i = 0\}}{\sum_{i=1}^{n} h(X_i, Y_i)\mathbf{1}\{A_i = 0\}}.$$

Note that $\widehat{F}_n(\cdot)$ and $\widehat{G}_n(\cdot)$'s are random functions serving as approximations to $F(\cdot)$ and $G(\cdot)$'s. Consider the Lagrangian functions

$$L(\theta, \lambda) = F(\theta) + \lambda G(\theta) \quad \text{and} \quad \widehat{L}_n(\theta, \lambda) = \widehat{F}_n(\theta) + \lambda \widehat{G}_n(\theta)$$

of the programs $(\mathcal{P}_0)$ and $(\mathcal{P}_n)$ respectively.

**Lemma C.1** (A version of Theorem 6.6.2 in Rubinstein & Shapiro (1993)). *Suppose that:*

(i) *The functions $F(\theta)$ and $G(\theta)$ are twice continuously differentiable.*

(ii) *The true program $(\mathcal{P}_0)$ has a unique optimal solution $\theta^\star$ and a unique Lagrange multiplier $\lambda^\star$ with $\theta^\star$ being an interior point of $\Theta$.*

(iii) *The Hessian matrix $\nabla^2 L(\theta^\star, \lambda^\star)$ is positive definite.*

(iv) *The random functions $\widehat{G}_n(\theta), k \in [K]$, are Lipschitz continuous in a neighborhood of $\theta^\star$ and differentiable at $\theta^\star$ with probability 1.*

(v)
$$\|\Delta_{in}(\theta^\star)\|_2 = O_p(n^{-1/2}), \quad i = 1, 2, 3$$

*and there is a neighborhood $U$ of $\theta^\star$ such that*

$$\sup_{\theta \in U} \frac{\|\Delta_{in}(\theta) - \Delta_{in}(\theta^\star)\|_2}{n^{-1/2} + \|\theta - \theta^\star\|_2} = o_p(1), \quad i = 1, 2, 3.$$

*Here we define random mappings $\Delta_{1n}(\theta) = \nabla\widehat{F}_n(\theta) - \nabla F(\theta)$, $\Delta_{2n}(\theta) = \widehat{G}_n(\theta) - G(\theta)$, and $\Delta_{3n}(\theta) = \nabla\widehat{G}_n(\theta) - \nabla G(\theta)$.*

(vi) *Random vectors $\sqrt{n}(\nabla\widehat{L}_n(\theta^\star, \lambda^\star), \widehat{G}_n(\theta^\star))$ converge in distribution to $\mathbf{Y} = (\mathbf{Y}_1, Y_2)$ as $n \to \infty$, where $\mathbf{Y}_1$ is a random vector and $Y_2$ is a random variable.*

Let $\widehat{\theta}_n$ be an optimal solution of $(\mathcal{P}_n)$ converging in probability as $n \to \infty$ to $\theta^\star$. Then

$$\sqrt{n}(\widehat{\theta}_n - \theta^\star) \xrightarrow{d} \bar{\boldsymbol{x}}(\boldsymbol{Y})$$

where $\bar{\boldsymbol{x}} = \bar{\boldsymbol{x}}(\boldsymbol{Y})$ is the optimal solution to the quadratic programming problem

$$\begin{array}{ll}
\underset{\boldsymbol{x}}{\text{minimize}} & \boldsymbol{x}^\top \boldsymbol{Y}_1 + \frac{1}{2}\boldsymbol{x}^\top \nabla^2 L(\theta^\star, \lambda^\star)\boldsymbol{x} \\
\text{subject to} & \nabla G(\theta^\star)^\top \boldsymbol{x} + Y_2 = 0
\end{array}.$$

Recall the standing assumptions, *(i)*, *(iv)*, *(v)* are guaranteed by the smoothness and concentration assumption, *(ii)* is postulated by the uniqueness assumption, and *(iii)* is made by our assumption. Now we derive the limiting distribution of random vectors $\sqrt{n}(\nabla \widehat{L}_n(\theta^\star, \lambda^\star), \widehat{G}_n(\theta^\star))$ required in *(vi)*.

For $a \in \{0, 1\}$, we have

$$\mathbb{E}\big[g(\theta^\star; X, Y)\mathbf{1}\{A = a\}\big] = \mathbb{P}(A = a)\mathbb{E}\big[g(\theta^\star; X, Y)|A = a\big] = \pi_a \mathbb{E}_{Q_a}[g],$$

and

$$\begin{aligned}
\text{Var}[g(\theta^\star; X, Y)\mathbf{1}\{A = a\}] &= \mathbb{E}\big[g^2(\theta^\star; X, Y)\mathbf{1}\{A = a\}\big] - \big\{\mathbb{E}\big[g(\theta^\star; X, Y)\mathbf{1}\{A = a\}\big]\big\}^2 \\
&= \pi_a \mathbb{E}_{Q_a}[g^2] - \pi_a^2 (\mathbb{E}_{Q_a}[g])^2 \\
&= \pi_a (\mathbb{E}_{Q_a}[g^2] - (\mathbb{E}_{Q_a}[g])^2) + (\pi_a - \pi_a^2)(\mathbb{E}_{Q_a}[g])^2 \\
&= \pi_a \text{Var}_{Q_a}[g] + \pi_0\pi_1(\mathbb{E}_{Q_a}[g])^2.
\end{aligned}$$

Similarly, for $a \in \{0, 1\}$, we have

$$\mathbb{E}\big[h(X, Y)\mathbf{1}\{A = a\}\big] = \pi_a \mathbb{E}_{Q_a}[h] \quad \text{and} \quad \text{Var}[h(X, Y)\mathbf{1}\{A = a\}] = \pi_a \text{Var}_{Q_a}[h] + \pi_0\pi_1(\mathbb{E}_{Q_a}[h])^2.$$

Moreover, we have

$$\begin{aligned}
&\text{Cov}(g(\theta^\star; X, Y)\mathbf{1}\{A = 1\}, g(\theta^\star; X, Y)\mathbf{1}\{A = 0\}) \\
=&\mathbb{E}\big[g^2(\theta^\star; X, Y)\mathbf{1}\{A = 0\}\mathbf{1}\{A = 1\}\big] - \mathbb{E}\big[g(\theta^\star; X, Y)\mathbf{1}\{A = 0\}\big] \times \mathbb{E}\big[g(\theta^\star; X, Y)\mathbf{1}\{A = 1\}\big] \\
=& - \pi_0\pi_1 \mathbb{E}_{Q_0}[g]\mathbb{E}_{Q_1}[g]
\end{aligned}$$

and similarly we can derive

$$\text{Cov}(h(X, Y)\mathbf{1}\{A = 1\}, h(X, Y)\mathbf{1}\{A = 0\}) = -\pi_0\pi_1 \mathbb{E}_{Q_0}[h]\mathbb{E}_{Q_1}[h],$$

$$\text{Cov}(g(\theta^\star; X, Y)\mathbf{1}\{A = a\}, h(X, Y)\mathbf{1}\{A = a\}) = \pi_a \text{Cov}_{Q_a}[g, h] + \pi_0\pi_1 \mathbb{E}_{Q_a}[g]\mathbb{E}_{Q_a}[h]$$

and

$$\text{Cov}(g(\theta^\star; X, Y)\mathbf{1}\{A = a\}, h(X, Y)\mathbf{1}\{A = 1 - a\}) = -\pi_0\pi_1 \mathbb{E}_{Q_a}[g]\mathbb{E}_{Q_{1-a}}[h]$$

for $a \in \{0, 1\}$.

Let $\boldsymbol{\eta}_1 = \mathbb{E}\big[\nabla \ell(\theta^\star; X, Y)\big]$, $\boldsymbol{\eta}_2 = \pi_1 \mathbb{E}_{Q_1}\big[\nabla g(\theta^\star; X, Y)\big]$ and $\boldsymbol{\eta}_3 = \pi_0 \mathbb{E}_{Q_0}\big[\nabla g(\theta^\star; X, Y)\big]$. By central limit theorem,

$$\sqrt{n}\left\{\begin{bmatrix}
n^{-1}\sum_{i=1}^n \nabla \ell(\theta^\star; X_i, Y_i) \\
n^{-1}\sum_{i=1}^n \nabla g(\theta^\star; X_i, Y_i)\mathbf{1}\{A_i = 1\} \\
n^{-1}\sum_{i=1}^n \nabla g(\theta^\star; X_i, Y_i)\mathbf{1}\{A_i = 0\} \\
n^{-1}\sum_{i=1}^n g(\theta^\star; X_i, Y_i)\mathbf{1}\{A_i = 1\} \\
n^{-1}\sum_{i=1}^n g(\theta^\star; X_i, Y_i)\mathbf{1}\{A_i = 0\} \\
n^{-1}\sum_{i=1}^n h(X_i, Y_i)\mathbf{1}\{A_i = 1\} \\
n^{-1}\sum_{i=1}^n h(X_i, Y_i)\mathbf{1}\{A_i = 0\}
\end{bmatrix} - \begin{bmatrix}
\boldsymbol{\eta}_1 \\
\boldsymbol{\eta}_2 \\
\boldsymbol{\eta}_3 \\
\pi_1 \mathbb{E}_{Q_1}[g] \\
\pi_0 \mathbb{E}_{Q_0}[g] \\
\pi_1 \mathbb{E}_{Q_1}[h] \\
\pi_0 \mathbb{E}_{Q_0}[h]
\end{bmatrix}\right\} \xrightarrow{d} \mathcal{N}\left(\mathbf{0}, \begin{bmatrix} \Omega_{11} & \Omega_{12} \\ \Omega_{21} & \Omega_{22} \end{bmatrix}\right),$$

(C.1)

where $\Omega_{11} \in \mathbb{R}^{3d \times 3d}, \Omega_{21} \in \mathbb{R}^{4 \times 3d}, \Omega_{12} = \Omega_{21}^\top, \Omega_{22}$ is given by

$$\begin{bmatrix}
\pi_1 Q_1^2[g] + \pi_0\pi_1(Q_1 g)^2 & -\pi_0\pi_1 Q_0 g Q_1 g & \pi_1 Q_1^2[g, h] + \pi_0\pi_1 Q_1 g Q_1 h & -\pi_0\pi_1 Q_0 h Q_1 g \\
-\pi_0\pi_1 Q_0 g Q_1 g & \pi_0 Q_0^2[g] + \pi_0\pi_1(Q_0 g)^2 & -\pi_0\pi_1 Q_0 g Q_1 h & \pi_0 Q_0^2[g, h] + \pi_0\pi_1 Q_0 g Q_0 h \\
\pi_1 Q_1^2[g, h] + \pi_0\pi_1 Q_1 g Q_1 h & -\pi_0\pi_1 Q_0 g Q_1 h & \pi_1 Q_1^2[h] + \pi_0\pi_1(Q_1 h)^2 & -\pi_0\pi_1 Q_0 h Q_1 h \\
-\pi_0\pi_1 Q_0 h Q_1 g & \pi_0 Q_0^2[g, h] + \pi_0\pi_1 Q_0 g Q_0 h & -\pi_0\pi_1 Q_0 h Q_1 h & \pi_0 Q_0^2[h] + \pi_0\pi_1(Q_0 h)^2
\end{bmatrix}.$$

Let function $w : \mathbb{R}^d \times \mathbb{R}^d \times \mathbb{R}^d \times \mathbb{R} \times \mathbb{R} \times \mathbb{R} \times \mathbb{R} \to \mathbb{R}^{d+1}$ be

$$w(\boldsymbol{v}_1, \boldsymbol{v}_2, \boldsymbol{v}_3, s_1, s_2, s_3, s_4) = \left(\boldsymbol{v}_1 + \lambda^\star \left\{\frac{\boldsymbol{v}_2}{s_3} - \frac{\boldsymbol{v}_3}{s_4}\right\}, \frac{s_1}{s_3} - \frac{s_2}{s_4}\right)^\top.$$

The gradient of function $w$ evaluated at

$$(\boldsymbol{v}_1, \boldsymbol{v}_2, \boldsymbol{v}_3) = (\boldsymbol{\eta}_1, \boldsymbol{\eta}_2, \boldsymbol{\eta}_3) \quad \text{and} \quad (s_1, s_2, s_3, s_4) = (\pi_1 \mathbb{E}_{Q_1}[g], \pi_0 \mathbb{E}_{Q_0}[g], \pi_1 \mathbb{E}_{Q_1}[h], \pi_0 \mathbb{E}_{Q_0}[h])$$

is given by

$$\nabla w = \begin{bmatrix} *_{3d \times d} & \mathbf{0}_{3d \times 1} \\ *_{4 \times d} & \xi_{4 \times 1} \end{bmatrix} \in \mathbb{R}^{(3d+4) \times (d+1)}$$

where

$$\xi = \left(\frac{1}{\pi_1 Q_1 h}, -\frac{1}{\pi_0 Q_0 h}, -\frac{Q_1 g}{\pi_1 (Q_1 h)^2}, \frac{Q_0 g}{\pi_0 (Q_0 h)^2}\right)^\top.$$

Applying delta method to (C.1) with $w(\cdot)$, we have

$$\sqrt{n} \begin{bmatrix} \nabla \widehat{L}_n(\theta^\star, \lambda^\star) \\ \widehat{G}_n(\theta^\star) \end{bmatrix} \xrightarrow{d} \mathcal{N}\left(\mathbf{0}, \nabla w^\top \begin{bmatrix} \Omega_{11} & \Omega_{12} \\ \Omega_{21} & \Omega_{22} \end{bmatrix} \nabla w\right) \xlongequal{d} \mathcal{N}\left(\mathbf{0}, \begin{bmatrix} \Sigma_{11} & \Sigma_{12} \\ \Sigma_{21} & \sigma^2 \end{bmatrix}\right),$$

where

$$\sigma^2 = \xi^\top \Omega_{22} \xi = \frac{Q_0^2[g]}{\pi_0 (Q_0 h)^3} + \frac{Q_0^2[h](Q_0 g)^2}{\pi_0 (Q_0 h)^4} - \frac{2Q_0^2[g, h] Q_0 g}{\pi_0 (Q_0 h)^3} + \frac{Q_1^2[g]}{\pi_1 (Q_1 h)^3} + \frac{Q_1^2[h](Q_1 g)^2}{\pi_1 (Q_1 h)^4} - \frac{2Q_1^2[g, h] Q_1 g}{\pi_1 (Q_1 h)^3} \tag{C.2}$$

Note that KKT condition implies

$$\boldsymbol{\eta}_1 + \lambda^\star \left\{\frac{\boldsymbol{\eta}_2}{\pi_1 Q_1 g} - \frac{\boldsymbol{\eta}_3}{\pi_0 Q_0 g}\right\} = 0 \quad \text{and} \quad \frac{Q_1 g}{Q_1 h} = \frac{Q_0 g}{Q_0 h} \triangleq \kappa. \tag{C.3}$$

Combining (C.2) and (C.3), we have

$$\begin{aligned}
&\sigma^2 \\
&= \frac{\mathrm{Var}_{Q_0}[g] + \mathrm{Var}_{Q_0}[\kappa h] - 2\,\mathrm{Cov}_{Q_0}[g, \kappa h]}{\pi_0 (\mathbb{E}_{Q_0}[h])^2} + \frac{\mathrm{Var}_{Q_1}[g] + \mathrm{Var}_{Q_1}[\kappa h] - 2\,\mathrm{Cov}_{Q_1}[g, \kappa h]}{\pi_1 (\mathbb{E}_{Q_1}[h])^2} \\
&= \frac{\mathrm{Var}_{Q_0}[g - \kappa h]}{\pi_0 (\mathbb{E}_{Q_0}[h])^2} + \frac{\mathrm{Var}_{Q_1}[g - \kappa h]}{\pi_1 (\mathbb{E}_{Q_1}[h])^2}.
\end{aligned} \tag{C.4}$$

Therefore, we conclude that the limiting distribution of $\sqrt{n}(\nabla \widehat{L}_n(\theta^\star, \lambda^\star), G_n(\theta^\star))$ is

$$(\boldsymbol{Y}_1, Y_2) \sim \mathcal{N}\left(\mathbf{0}, \begin{bmatrix} \Sigma_{11} & \Sigma_{12} \\ \Sigma_{21} & \sigma^2 \end{bmatrix}\right).$$

By Lemma (C.1), we have

$$\sqrt{n}(\widehat{\theta}_n - \theta^\star) \xrightarrow{d} \bar{\boldsymbol{x}},$$

where $\bar{\boldsymbol{x}}$ is given by the linear system

$$\underbrace{\begin{bmatrix} \nabla^2 L(\theta^\star, \lambda^\star) & \nabla G(\theta^\star) \\ \nabla G(\theta^\star)^\top & 0 \end{bmatrix}}_{\triangleq B} \begin{bmatrix} \bar{\boldsymbol{x}} \\ \bar{\lambda} \end{bmatrix} = -\begin{bmatrix} \boldsymbol{Y}_1 \\ Y_2 \end{bmatrix} \sim \mathcal{N}\left(\mathbf{0}, \begin{bmatrix} \Sigma_{11} & \Sigma_{12} \\ \Sigma_{21} & \sigma^2 \end{bmatrix}\right),$$

or

$$\begin{bmatrix} \bar{\boldsymbol{x}} \\ \bar{\lambda} \end{bmatrix} \sim \mathcal{N}\left(\mathbf{0}, B^{-1} \begin{bmatrix} \Sigma_{11} & \Sigma_{12} \\ \Sigma_{21} & \sigma^2 \end{bmatrix} B^{-1}\right), \tag{C.5}$$

which implies $\sqrt{n}(\widehat{\theta}_n - \theta^\star) \xrightarrow{d} \bar{\boldsymbol{x}} \sim \mathcal{N}(\mathbf{0}, \bar{\Sigma})$ for some $\bar{\mu}$ and $\bar{\Sigma}$ determined by (C.5).

By delta method, we have

$$\sqrt{n}G(\widehat{\theta}_n) = \sqrt{n}\{G(\widehat{\theta}_n) - \underbrace{G(\theta^\star)}_{=0}\} \xrightarrow{d} \mathcal{N}(0, \nabla G(\theta^\star)^\top \bar{\Sigma} \nabla G(\theta^\star)).$$

Now we calculate $\nabla G(\theta^\star)^\top \bar{\Sigma} \nabla G(\theta^\star)$.

For notation simplicity, we denote $\nabla^2 L = \nabla^2 L(\theta^\star, \lambda^\star), \nabla G = \nabla G(\theta^\star)$ and $H = (\nabla^2 L)^{-1}\nabla G[\nabla G^\top (\nabla^2 L)^{-1}\nabla G]^{-1}$. By block matrix inversion, we have

$$B^{-1} = \begin{bmatrix} (\nabla^2 L)^{-1} - H\nabla G^\top (\nabla^2 L)^{-1} & H \\ H^\top & -[\nabla G^\top (\nabla^2 L)^{-1}\nabla G]^{-1}. \end{bmatrix}$$

Note that $\nabla G^\top H = 1$ and $\nabla G^\top \left\{(\nabla^2 L)^{-1} - H\nabla G^\top (\nabla^2 L)^{-1}\right\} = 0$. We have

$$\nabla G(\theta^\star)^\top \bar{\Sigma} \nabla G(\theta^\star)$$
$$=\nabla G^\top \left[\left\{(\nabla^2 L)^{-1} - H\nabla G^\top (\nabla^2 L)^{-1}\right\}\Sigma_{11} + H\Sigma_{21}\right] \underbrace{\left\{(\nabla^2 L)^{-1} - (\nabla^2 L)^{-1}\nabla G H^\top\right\}\nabla G}_{=0}$$
$$+ \underbrace{\nabla G^\top \left\{(\nabla^2 L)^{-1} - H\nabla G^\top (\nabla^2 L)^{-1}\right\}}_{=0}\Sigma_{12}H^\top \nabla G + \nabla G^\top H\sigma^2 H^\top \nabla G$$
$$=\sigma^2.$$

Therefore, we conclude that

$$\sqrt{n}\{c(\widehat{\theta}_n) - \cancel{c(\theta^\star)}\} = \sqrt{n}G(\widehat{\theta}_n)$$
$$\xrightarrow{d} \mathcal{N}(0, \sigma^2) \overset{d}{=} \mathcal{N}\left(0, \frac{\mathrm{Var}_{Q_0}[g - \kappa h]}{\pi_0(\mathbb{E}_{Q_0}[h])^2} + \frac{\mathrm{Var}_{Q_1}[g - \kappa h]}{\pi_1(\mathbb{E}_{Q_1}[h])^2}\right).$$

By a similar argument, we have

$$\sqrt{n}\{\psi_{\mathrm{frac}} \times c(\widetilde{\theta}_n) - \cancel{\psi_{\mathrm{frac}} \times c(\theta^\star)}\} \xrightarrow{d} \mathcal{N}\left(0, \frac{\mathrm{Var}_{\widetilde{Q}_0}[g - \kappa h]}{\widetilde{\pi}_0(\mathbb{E}_{\widetilde{Q}_0}[h])^2} + \frac{\mathrm{Var}_{\widetilde{Q}_1}[g - \kappa h]}{\widetilde{\pi}_1(\mathbb{E}_{\widetilde{Q}_1}[h])^2}\right),$$

which implies

$$\sqrt{n} \times c(\widetilde{\theta}_n) \xrightarrow{d} \mathcal{N}(0, \widetilde{\sigma}^2) \overset{d}{=} \mathcal{N}\left(0, \psi_{\mathrm{frac}}^{-2} \times \left\{\frac{\mathrm{Var}_{\widetilde{Q}_0}[g - \kappa h]}{\widetilde{\pi}_0(\mathbb{E}_{\widetilde{Q}_0}[h])^2} + \frac{\mathrm{Var}_{\widetilde{Q}_1}[g - \kappa h]}{\widetilde{\pi}_1(\mathbb{E}_{\widetilde{Q}_1}[h])^2}\right\}\right).$$

Now, we prove the case when $\alpha_n = o(\frac{1}{\sqrt{n}})$. For this case note that the equality constraint for the population problem can be rewritten as two inequality constraints:

$$(\mathcal{P}_0): \quad \theta^\star \in \begin{cases} \arg\min_{\theta \in \Theta} & \mathbb{E}\big[\ell(\theta; X, Y)\big] \\ \text{subject to} & \dfrac{\mathbb{E}\big[g(\theta; X, Y)|A = 1\big]}{\mathbb{E}\big[h(X, Y)|A = 1\big]} - \dfrac{\mathbb{E}\big[g(\theta; X, Y)|A = 0\big]}{\mathbb{E}\big[h(X, Y)|A = 0\big]} \leq 0 \\ & \dfrac{\mathbb{E}\big[g(\theta; X, Y)|A = 0\big]}{\mathbb{E}\big[h(X, Y)|A = 0\big]} - \dfrac{\mathbb{E}\big[g(\theta; X, Y)|A = 1\big]}{\mathbb{E}\big[h(X, Y)|A = 1\big]} \leq 0 \end{cases},$$

where the expectation is with respect to the underlying distribution of tuple $(X, A, Y)$.

The corresponding empirical problem given the true sensitive attribute is

$$(\mathcal{P}_n): \quad \widehat{\theta}_n \in \begin{cases} \arg\min_{\theta \in \Theta} & \dfrac{1}{n}\sum_{i=1}^n \ell(\theta; X_i, Y_i) \\ \text{subject to} & \dfrac{\sum_{i=1}^n g(\theta; X_i, Y_i)\mathbf{1}\{A_i = 1\}}{\sum_{i=1}^n h(X_i, Y_i)\mathbf{1}\{A_i = 1\}} - \dfrac{\sum_{i=1}^n g(\theta; X_i, Y_i)\mathbf{1}\{A_i = 0\}}{\sum_{i=1}^n h(X_i, Y_i)\mathbf{1}\{A_i = 0\}} - \alpha_n \leq 0 \\ & \dfrac{\sum_{i=1}^n g(\theta; X_i, Y_i)\mathbf{1}\{A_i = 0\}}{\sum_{i=1}^n h(X_i, Y_i)\mathbf{1}\{A_i = 0\}} - \dfrac{\sum_{i=1}^n g(\theta; X_i, Y_i)\mathbf{1}\{A_i = 1\}}{\sum_{i=1}^n h(X_i, Y_i)\mathbf{1}\{A_i = 1\}} - \alpha_n \leq 0 \end{cases}.$$

The corresponding empirical problem given the proxy sensitive attribute is

$$(\widetilde{\mathcal{P}}_n): \quad \widetilde{\theta}_n \in \begin{cases} \arg\min_{\theta \in \Theta} & \frac{1}{n}\sum_{i=1}^{n} \ell(\theta; X_i, Y_i) \\ \text{subject to} & \frac{\sum_{i=1}^{n} g(\theta; X_i, Y_i)\mathbf{1}\{Z_i=1\}}{\sum_{i=1}^{n} h(X_i, Y_i)\mathbf{1}\{Z_i=1\}} - \frac{\sum_{i=1}^{n} g(\theta; X_i, Y_i)\mathbf{1}\{Z_i=0\}}{\sum_{i=1}^{n} h(X_i, Y_i)\mathbf{1}\{Z_i=0\}} - \alpha_n \leq 0 \\ & \frac{\sum_{i=1}^{n} g(\theta; X_i, Y_i)\mathbf{1}\{Z_i=0\}}{\sum_{i=1}^{n} h(X_i, Y_i)\mathbf{1}\{Z_i=0\}} - \frac{\sum_{i=1}^{n} g(\theta; X_i, Y_i)\mathbf{1}\{Z_i=1\}}{\sum_{i=1}^{n} h(X_i, Y_i)\mathbf{1}\{Z_i=1\}} - \alpha_n \leq 0 \end{cases}.$$

We denote

$$F(\theta) = \mathbb{E}\big[\ell(\theta; X, Y)\big], \widehat{F}_n(\theta) = \frac{1}{n}\sum_{i=1}^{n} \ell(\theta; X_i, Y_i),$$

$$G_1(\theta) = \frac{\mathbb{E}\big[g(\theta; X, Y)|A=1\big]}{\mathbb{E}\big[h(X, Y)|A=1\big]} - \frac{\mathbb{E}\big[g(\theta; X, Y)|A=0\big]}{\mathbb{E}\big[h(X, Y)|A=0\big]},$$

$$G_2(\theta) = \frac{\mathbb{E}\big[g(\theta; X, Y)|A=0\big]}{\mathbb{E}\big[h(X, Y)|A=0\big]} - \frac{\mathbb{E}\big[g(\theta; X, Y)|A=1\big]}{\mathbb{E}\big[h(X, Y)|A=1\big]},$$

$$\widehat{G}_{1n}(\theta) = \frac{\sum_{i=1}^{n} g(\theta; X_i, Y_i)\mathbf{1}\{A_i=1\}}{\sum_{i=1}^{n} h(X_i, Y_i)\mathbf{1}\{A_i=1\}} - \frac{\sum_{i=1}^{n} g(\theta; X_i, Y_i)\mathbf{1}\{A_i=0\}}{\sum_{i=1}^{n} h(X_i, Y_i)\mathbf{1}\{A_i=0\}} - \alpha_n,$$

and

$$\widehat{G}_{2n}(\theta) = \frac{\sum_{i=1}^{n} g(\theta; X_i, Y_i)\mathbf{1}\{A_i=0\}}{\sum_{i=1}^{n} h(X_i, Y_i)\mathbf{1}\{A_i=0\}} - \frac{\sum_{i=1}^{n} g(\theta; X_i, Y_i)\mathbf{1}\{A_i=1\}}{\sum_{i=1}^{n} h(X_i, Y_i)\mathbf{1}\{A_i=1\}} - \alpha_n.$$

Consider the Lagrangian functions

$$L(\theta, \boldsymbol{\lambda}) = F(\theta) + \lambda_1 G_1(\theta) + \lambda_2 G_2(\theta) \quad \text{and} \quad \widehat{L}_n(\theta, \boldsymbol{\lambda}) = \widehat{F}_n(\theta) + \lambda_1 \widehat{G}_{1n}(\theta) + \lambda_2 \widehat{G}_{2n}(\theta).$$

of the programs $(\mathcal{P}_0)$ and $(\mathcal{P}_n)$ respectively.

**Lemma C.2** (A version of Theorem 6.6.2 in Rubinstein & Shapiro (1993)). *Suppose that:*

- *(i) The functions $F(\theta)$, $G_1(\theta)$ and $G_2(\theta)$ are twice continuously differentiable.*
- *(ii) The true program $(\mathcal{P}_0)$ has a unique optimal solution $\theta^\star$ and a unique Lagrange multiplier $\boldsymbol{\lambda}^\star$ with $\theta^\star$ being an interior point of $\Theta$.*
- *(iii) The Hessian matrix $\nabla^2 L(\theta^\star, \boldsymbol{\lambda}^\star)$ is positive definite.*
- *(iv) The random functions $\widehat{G}_{1n}(\theta)$ and $\widehat{G}_{2n}(\theta), k \in [K]$, are Lipschitz continuous in a neighborhood of $\theta^\star$ and differentiable at $\theta^\star$ with probability 1.*
- *(v)*

$$\|\Delta_{in}(\theta^\star)\|_2 = O_p(n^{-1/2}), \quad i=1,2,3$$

*and there is a neighborhood $U$ of $\theta^\star$ such that*

$$\sup_{\theta \in U} \frac{\|\Delta_{in}(\theta) - \Delta_{in}(\theta^\star)\|_2}{n^{-1/2} + \|\theta - \theta^\star\|_2} = o_p(1), \quad i=1,2,3.$$

*Here we define random mappings $\Delta_{1n}(\theta) = \nabla\widehat{F}_n(\theta) - \nabla F(\theta)$, $\Delta_{2n}(\theta) = \widehat{G}_n(\theta) - G(\theta)$, and $\Delta_{3n}(\theta) = \nabla\widehat{G}_n(\theta) - \nabla G(\theta)$.*
- *(vi) Random vectors $\sqrt{n}(\nabla\widehat{L}_n(\theta^\star, \boldsymbol{\lambda}^\star), \widehat{G}_{1n}(\theta^\star)), \widehat{G}_{2n}(\theta^\star))$ converge in distribution to $\boldsymbol{Y} = (Y_1, Y_2, Y_3)$ as $n \to \infty$, where $Y_1$ is a random vector and $Y_2$ and $Y_3$ are random variables.*

*Let $\widehat{\theta}_n$ be an optimal solution of $(\mathcal{P}_n)$ converging in probability as $n \to \infty$ to $\theta^\star$. Then*

$$\sqrt{n}(\widehat{\theta}_n - \theta^\star) \xrightarrow{d} \bar{\boldsymbol{x}}(\boldsymbol{Y})$$

*where $\bar{\boldsymbol{x}} = \bar{\boldsymbol{x}}(\boldsymbol{Y})$ is the optimal solution to the quadratic programming problem*

$$\begin{aligned} \underset{\boldsymbol{x}}{\text{minimize}} \quad & \boldsymbol{x}^\top \boldsymbol{Y}_1 + \frac{1}{2}\boldsymbol{x}^\top \nabla^2 L(\theta^\star, \boldsymbol{\lambda}^\star)\boldsymbol{x} \\ \text{subject to} \quad & \nabla G_1(\theta^\star)^\top \boldsymbol{x} + Y_2 \leq 0 \\ & \nabla G_2(\theta^\star)^\top \boldsymbol{x} + Y_3 \leq 0 \end{aligned} \quad .$$

Note that

$$\nabla G_1(\theta^\star)^\top \boldsymbol{x} + Y_2 \leq 0 \iff \nabla G(\theta^\star)^\top \boldsymbol{x} + Y \leq 0$$

and

$$\nabla G_1(\theta^\star)^\top \boldsymbol{x} + Y_2 \leq 0 \iff -\nabla G(\theta^\star)^\top \boldsymbol{x} + (-Y) \leq 0.$$

Therefore the last quadratic programming problem with two inequality constraints reduces to the quadratic programming problem with single equality constraint when $\alpha_n \equiv 0$. The limiting distributional results thus persist as we proved for the $\alpha_n \equiv 0$ case.

Lastly, we calculate the asymptotic relative efficiency (ARE) of $\widetilde{\theta}_n$ to $\widehat{\theta}_n$.

Recall that

$$\sigma^2 = \frac{\mathrm{Var}_{Q_0}[g - \kappa h]}{\pi_0(\mathbb{E}_{Q_0}[h])^2} + \frac{\mathrm{Var}_{Q_1}[g - \kappa h]}{\pi_1(\mathbb{E}_{Q_1}[h])^2},$$

$$\widetilde{\sigma}^2 = \psi_{\mathrm{frac}}^{-2} \times \left\{ \frac{\mathrm{Var}_{\widetilde{Q}_0}[g - \kappa h]}{\widetilde{\pi}_0(\mathbb{E}_{\widetilde{Q}_0}[h])^2} + \frac{\mathrm{Var}_{\widetilde{Q}_1}[g - \kappa h]}{\widetilde{\pi}_1(\mathbb{E}_{\widetilde{Q}_1}[h])^2} \right\}$$

$$= \psi_{\mathrm{frac}}^{-2} \times \left\{ \frac{(1-\gamma)\pi_0 \,\mathrm{Var}_{Q_0}[g - \kappa h] + \gamma\pi_1 \,\mathrm{Var}_{Q_1}[g - \kappa h]}{\{(1-\gamma)\pi_0\mathbb{E}_{Q_0}[h] + \gamma\pi_1\mathbb{E}_{Q_1}[h]\}^2} \right.$$

$$\left. + \frac{\gamma\pi_0 \,\mathrm{Var}_{Q_0}[g - \kappa h] + (1-\gamma)\pi_1 \,\mathrm{Var}_{Q_1}[g - \kappa h]}{\{\gamma\pi_0\mathbb{E}_{Q_0}[h] + (1-\gamma)\pi_1\mathbb{E}_{Q_1}[h]\}^2} \right\},$$

and

$$\psi_{\mathrm{frac}} = \frac{(1-2\gamma)\pi_0\pi_1\mathbb{E}_{Q_0}[h]\mathbb{E}_{Q_1}[h]}{\{\gamma\pi_0\mathbb{E}_{Q_0}[h] + (1-\gamma)\pi_1\mathbb{E}_{Q_1}[h]\}\{(1-\gamma)\pi_0\mathbb{E}_{Q_0}[h] + \gamma\pi_1\mathbb{E}_{Q_1}[h]\}}.$$

Therefore, we have

$$\mathrm{ARE}(\widetilde{\theta}_n, \widehat{\theta}_n) = \frac{\sigma^2}{\widetilde{\sigma}^2} = \varphi\left( \gamma, \frac{\pi_0\mathbb{E}_{Q_0}[h]}{\pi_1\mathbb{E}_{Q_1}[h]}, \frac{\mathrm{Var}_{Q_0}[g(\theta^\star; X, Y) - \kappa h(X, Y)]/\mathbb{E}_{Q_0}[h]}{\mathrm{Var}_{Q_1}[g(\theta^\star; X, Y) - \kappa h(X, Y)]/\mathbb{E}_{Q_1}[h]} \right)$$

$$= \varphi\left( \gamma, \frac{\pi_0 m_0}{\pi_1 m_1}, \frac{\mathrm{Var}[g(\theta^\star; X, Y) - \kappa h(X, Y)|A = 0]/m_0}{\mathrm{Var}[g(\theta^\star; X, Y) - \kappa h(X, Y)|A = 1]/m_1} \right),$$

where

$$\varphi(\gamma, r_1, r_2) \triangleq \frac{(1-2\gamma)^2 r_1(r_1 + r_2)}{\{\gamma r_1 + (1-\gamma)\}^2\{(1-\gamma)r_1 r_2 + \gamma\} + \{(1-\gamma)r_1 + \gamma\}^2\{\gamma r_1 r_2 + (1-\gamma)\}}.$$

Hence we complete the proof of Theorem 3.1. $\qquad\square$

## D   MULTIPLE DEMOGRAPHIC GROUPS

We provide further discussion to supplement Section 3.1.

Note that the fairness notion (3.5) uses group 0 as a reference group. One can also define a fairness notion by

$$\frac{\mathbb{E}\big[g(\theta; X, Y)|A = k\big]}{\mathbb{E}\big[h(X, Y)|A = k\big]} - \frac{\mathbb{E}\big[g(\theta; X, Y)\big]}{\mathbb{E}\big[h(X, Y)\big]} = 0 \quad \text{for } k \in \{0\} \cup [K] \tag{D.1}$$

which is symmetric in group indices. Due to the equivalence of (D.1) and (3.5), we opt to use (3.5) for a comparison with two-group theory.

Theorem 3.2 is a direct extension of Theorem 3.1 and follows the same proof procedure as of Theorem 3.1. Moreover, let $h \equiv 1$, the linear-fractional fairness (3.5) degenerates into linear fairness:

$$\mathbb{E}\big[g(\theta; X, Y)|A = k\big] - \mathbb{E}\big[g(\theta; X, Y)|A = 0\big] = 0 \quad \text{for } k \in [K]. \tag{D.2}$$

By Theorem 3.2, we immediately have the following corollary.

**Theorem D.1** (Privacy cost in linear fairness (D.2)-aware learning). *Under the standing assumptions, let estimators $\widehat{\theta}_n$ and $\widetilde{\theta}_n$ be consistent for $\theta^\star$, then*

$$\sqrt{n}\{c(\widehat{\theta}_n) - c(\theta^\star)\} \xrightarrow{d} \mathcal{N}(0, \Sigma) \text{ and } \sqrt{n}\{c(\widetilde{\theta}_n) - c(\theta^\star)\} \xrightarrow{d} \mathcal{N}(0, \Psi_{\text{lin}}^{-1}\widetilde{\Sigma}\Psi_{\text{lin}}^{-\top}),$$

*where*

$$\Sigma_{kl} = \frac{\text{Var}_{Q_0}[g(\theta^\star; X, Y)]}{\pi_0} + \left(\frac{\text{Var}_{Q_k}[g(\theta^\star; X, Y)]}{\pi_k}\right) \mathbf{1}\{k = l\}$$

$$\widetilde{\Sigma}_{kl} = \frac{\text{Var}_{\widetilde{Q}_0}[g(\theta^\star; X, Y)]}{\widetilde{\pi}_0} + \left(\frac{\text{Var}_{\widetilde{Q}_k}[g(\theta^\star; X, Y)]}{\widetilde{\pi}_k}\right) \mathbf{1}\{k = l\}$$

$$\Psi_{\text{lin}} = \begin{cases} \left(\frac{1 - K\gamma}{\widetilde{\pi}_k} - \frac{\gamma}{\widetilde{\pi}_0}\right)\pi_k & \text{if } k = l \\ \left(\frac{1}{\widetilde{\pi}_k} - \frac{1}{\widetilde{\pi}_0}\right)\gamma\pi_l & \text{if } k \neq l \end{cases}$$

*for $k, l \in [K]$.*

## E  MISSING SENSITIVE ATTRIBUTES

Under the missingness mechanism (3.8), the probability of observing a complete sample from group $a$ is

$$\mathbb{P}(A = a, R = 1) = \omega_a\pi_a$$

for $a \in \{0, 1\}$. By the intermediate conclusion of Theorem 3.1, we have

$$\sqrt{n}\{c(\widehat{\theta}_n) - c(\theta^\star)\} \xrightarrow{d} \mathcal{N}\left(0, \frac{\text{Var}_{Q_0}[g(\theta^\star; X, Y) - \kappa h(X, Y)]}{\pi_0(\mathbb{E}_{Q_0}[h(X, Y)])^2} + \frac{\text{Var}_{Q_1}[g(\theta^\star; X, Y) - \kappa h(X, Y)]}{\pi_1(\mathbb{E}_{Q_1}[h(X, Y)])^2}\right),$$

and

$$\sqrt{n}\{c(\widetilde{\theta}_n) - c(\theta^\star)\} \xrightarrow{d} \mathcal{N}\left(0, \frac{\text{Var}_{Q_0}[g(\theta^\star; X, Y) - \kappa h(X, Y)]}{\omega_0\pi_0(\mathbb{E}_{Q_0}[h(X, Y)])^2} + \frac{\text{Var}_{Q_1}[g(\theta^\star; X, Y) - \kappa h(X, Y)]}{\omega_1\pi_1(\mathbb{E}_{Q_1}[h(X, Y)])^2}\right).$$

Comparing the two asymptotic variances, we conclude that

$$\text{ARE}(\widetilde{\theta}_n, \widehat{\theta}_n) = \frac{r_2 + r_1}{\omega_0^{-1}r_2 + \omega_1^{-1}r_1},$$

where

$$r_1 = \frac{\pi_0 m_0}{\pi_1 m_1} \text{ and } r_2 = \frac{\text{Var}[g(\theta^\star; X, Y) - \kappa h(X, Y)|A = 0]/m_0}{\text{Var}[g(\theta^\star; X, Y) - \kappa h(X, Y)|A = 1]/m_1}.$$

