# OpenReview forum: "The Cost of Privacy in Fair Machine Learning"
_ICLR.cc/2023/Conference — Submitted to ICLR 2023_

### Official Review · Reviewer_YkN8 · 2022-10-12

**Confidence:** 4
**Correctness:** 4
**Technical Novelty And Significance:** 3
**Empirical Novelty And Significance:** 1
**Recommendation:** 3

**Clarity, Quality, Novelty And Reproducibility:**

This paper is well-written and the notations are clear. The paper is novel from a theoretical perspective, but it requires additional discussions regarding its setup. I don’t see any reproducibility issues.

**Strength And Weaknesses:**

The problems studied in this paper (i.e., characterizing the effect of privacy cost in terms of group fairness) are important and significant. This paper, including its appendix, is well-written and the notations are very clear. The theoretical results look solid.

My main concerns about this paper are listed below.

**Privacy mechanism**: it is unclear to me why the privacy mechanism is only applied to the sensitive attributes (S). When S is correlated with the rest of the features, an adversary can potentially reconstruct it from the rest of the features. Hence, I think it is crucial to apply the local DP mechanism to the entire data point instead of the sensitive attributes only.

**Usefulness and assumptions**: Group fairness constraints are often non-convex w.r.t. the model parameters. Hence, from a practical perspective, it is unclear to me that studying the consistency of the *global* optimal solutions of empirical risk minimization is meaningful. From a theoretical perspective, it is unclear to me how to validate the assumptions in this paper. For example, Assumption 2 requires the uniqueness of the optimal primal-dual solution. This assumption makes a lot of sense in convex analysis but once again group fairness constraints are often non-convex. I would love to see some examples to clarify these assumptions.

**Experiments**: The authors only validate their theoretical results on an artificial dataset. The authors may consider demonstrating their results through some real-world datasets (e.g., adult, compas). These datasets can also help validate the assumptions made in this paper.

Other comments:

**Fairness measures**: the authors may consider including a discussion on what group fairness measures can be covered by their analysis. For example, can the analysis cover false discovery rate and calibration error?

**Related work**: this paper is missing some references. For example, the problem of learning from noisy observations of (or missing) sensitive attributes have been studied in fair ML by a line of works and I think it is worth acknowledging these references in the paper.

**Intersectionality**: what if there are intersection sub-groups? Can the analysis in this paper extend to such settings?

**Summary Of The Paper:**

This paper considers the problem of learning a fair classifier from a sanitized dataset. This sanitized dataset is obtained by applying a local DP mechanism to the original data to obfuscate the sensitive attributes. This paper provides a precise characterization of the utility reduction in terms of statistical efficiency. The authors also consider several extensions of their theory (e.g., multiple groups, missing sensitive attributes). Simulations have been done on an artificial dataset, created by the authors.

**Summary Of The Review:**

Interesting paper but I have several concerns that require clarification and additional results

---

> ### Author Response · Authors · 2022-11-18
> **Response to Reviewer YkN8**
>
> We thank the reviewer for the valuable feedback. We address questions and concerns below.
>
> > It is unclear to me why the privacy mechanism is only applied to the sensitive attributes (S). When S is correlated with the rest of the features, an adversary can potentially reconstruct it from the rest of the features. Hence, I think it is crucial to apply the local DP mechanism to the entire data point instead of the sensitive attributes only.
>
> We analyze the statistical efficiency cost of privacy under Mozannar et al (2020)’s fair learning with private sensitive attributes framework. The reviewer's concern generates a new research question: what is the cost to statistical efficiency if we construct a noisy constraint using imputed sensitive attributes reconstructed from the features? We concur with the reviewer that this research problem is significant and intriguing; however, it does not fit our current mathematical tools and is, therefore, outside the scope of the present paper.
>
> > Group fairness constraints are often non-convex w.r.t. the model parameters. Hence, from a practical perspective, it is unclear to me that studying the consistency of the global optimal solutions of empirical risk minimization is meaningful.
>
> Our theory is founded on perturbation analysis of KKT points, so our results remain valid for non-convex local minima. That is to say, our results are valid so long as a series of empirical estimators converges to a population-level local minimum.
>
> > Fairness measures: the authors may consider including a discussion on what group fairness measures can be covered by their analysis. For example, can the analysis cover false discovery rate and calibration error?
>
> Our analysis covers for example equal opportunity, predictive equality, equalized odds, demographic parity, risk parity, true positive (true negative, false positive, false negative) rate parity, but doesn't cover false discovery rate and calibration error. Suppose a group fairness measure involves conditional probability $\mathbb{P}\\{\text{event}|\text{condition}\\}$, where the event and condition are associated with $(X, A, Y, \widehat{Y})$. If the condition does not involve $\widehat{Y}$, then our analysis will cover such a group fairness measure.
>
> > What if there are intersection sub-groups? Can the analysis in this paper extend to such settings?
>
> Our results can be extended to the setting of intersection sub-groups, which involves a more complicated form of asymptotic variances and asymptotic relative efficiency.

---

### Official Review · Reviewer_MtpY · 2022-10-23

**Confidence:** 4
**Correctness:** 4
**Technical Novelty And Significance:** 4
**Empirical Novelty And Significance:** 1
**Recommendation:** 5

**Clarity, Quality, Novelty And Reproducibility:**

- The paper is clearly written.
- To my knowledge, the ARE measure is original.

**Details Of Ethics Concerns:**

No concerns.

**Strength And Weaknesses:**

Strengths
- First or one of the first works to make a theoretical connection between a differentially-private sensitive attribute and model fairness convergence.
- The metric is statistically principled and provides interpretability.
- The mathematical derivations are straightforward.
- Simulations show a fairness-privacy trade-off as expected.

Weaknesses
- While ARE can be quite useful in connecting privacy and fairness, it is not clear how practically it can be applied. The authors state three technical assumptions - smoothness and concentration, uniqueness, and positive definiteness - without much justification other than they give interpretability. It is not clear how reasonable they are for various fairness applications. For example, if a company is using ML for hiring, how can it be confident that the properties hold without actually proving them? A good measure is one that everyone can use to evaluate any technique, so IMHO the applicability is critical.
- Another question is whether ARE is too complicated for its purpose. A simple approach would be to look at the $\epsilon$ value of differential privacy and fairness violation $c$ together to see the privacy-fairness tradeoff. There should be some justification through simulations or experiments on why much simpler approaches do not capture the tradeoff as well as ARE.
- Section 3.1 (General Theory) extends the two demographic groups theory to multiple groups, but the results look quite similar and thus redundant. How about discussing the general theory from the start and focus more on why the measure is practical by introducing various use cases? Also for the general theory, it is not clear if the same three technical assumptions are sufficient.
- The simulations are definitely helpful, but seem too thin to convincingly demonstrate ARE. Here are some potentially-interesting experiments that could strengthen the paper:
  - A comparison with some baselines that also capture the privacy-fairness tradeoff (e.g., any combinations of $\epsilon$ and $c$) to clearly see the benefits of ARE.
  - A demonstration that users actually think ARE is interpretable through user studies.
  - An experiment on real datasets.
  - An experiment that involves the constraint violation inflation.
  - An experiment that assumes missing sensitive attributes.
- In Section 5, there is some more analysis where $h \equiv 1$, but it is not clear if this belongs in the summary and discussion.




**Summary Of The Paper:**

This paper proposes a cost measure that connects privacy to model fairness. Given that sensitive attributes are private and protected using differential privacy, the main contribution is the introduction of asymptotic relative efficiency (ARE), which measures how much the convergence to a fair model is "slowed down" compared to when not injecting the noise by comparing the variances of estimators. The derivation is first performed in a binary sensitive group setup and then generalized to multiple groups. A simple simulation is performed followed by a discussion.

**Summary Of The Review:**

The paper proposes an interesting novel measure to quantify how differential privacy noise on a sensitive attribute influences model fairness convergence. While the mathematical derivation looks sound, it is not clear how widely applicable the measure is. In addition, the repetition of content between the binary versus general theories seems to be unnecessary. Finally, the experiments can be improved in several ways.

---

> ### Author Response · Authors · 2022-11-18
> **Response to Reviewer MtpY**
>
> We thank the reviewer for the valuable feedback. We address questions and concerns below.
>
> > While ARE can be quite useful in connecting privacy and fairness, it is not clear how practically it can be applied.
>
> ARE is a measure of constraint violations at test time. It is generally not possible to evaluate this metric in practice, but it is meaningful as a measure of the statistical efficiency of a learning algorithm. For example, estimation error or generalization error are both measures that are commonly used to measure the statistical efficiency of learning algorithms, but you cannot evaluate them in practice.
>
> > Another question is whether ARE is too complicated for its purpose. A simple approach would be to look at the $c$ value of differential privacy and fairness violation  together to see the privacy-fairness tradeoff. There should be some justification through simulations or experiments on why much simpler approaches do not capture the tradeoff as well as ARE.
>
> First, the value of fairness violation $c$ does not tell its variance. ARE compares the variances of constraint violation of estimators. Second, to determine an estimator's variance of fairness violation, one must repeatedly resample the dataset and retrain the model. ARE requires neither resampling nor retraining.
>
> > Section 3.1 (General Theory) extends the two demographic groups theory to multiple groups, but the results look quite similar and thus redundant. How about discussing the general theory from the start and focus more on why the measure is practical by introducing various use cases? Also for the general theory, it is not clear if the same three technical assumptions are sufficient.
>
> We begin with a two-group case to give the reader a better understanding of our analysis. In the revision, we will expand Section 2.4's discussion of the statistical efficiency loss measurement. The three technical assumptions are sufficient for us to apply perturbation analysis to the population-level minimum.
>
> > In Section 5, there is some more analysis where $h\equiv 1$, but it is not clear if this belongs in the summary and discussion.
>
> It belongs in the summary and discussion section. We use a simple case $h \equiv 1$ to describe a potential future research direction, namely, how can we improve the statistical efficiency under the current problem setup?

---

### Official Review · Reviewer_gE91 · 2022-10-24

**Confidence:** 3
**Correctness:** 4
**Technical Novelty And Significance:** 3
**Empirical Novelty And Significance:** Not applicable
**Recommendation:** 5

**Clarity, Quality, Novelty And Reproducibility:**

This paper can benefit from improvements in clarity:
1. Notations introduced through out the paper are not always explained (e.g. does the crossed-out operator mean the true mean? (Definition 2.5 and Theorem 3.1)
2. It is not clear what definitions the authors introduce for the first time and which definitions are cited from prior work. (e.g. is asymptotic relative efficiency an existing concept or newly introduced by the authors?)

No experiments on real data to reproduce. Simulated data only.


**Strength And Weaknesses:**

Strengths:
1. Extends analysis to the multi-group setting.
2. The authors include simulated experiments to demonstrate the privacy cost.
3. Characterizing the cost of protecting sensitive attributes can be useful in many real-world settings.

Weaknesses:
1. Implicit assumption that demographic features $A$ are independent of the $X$. In many real-life datasets, there is almost an exact correlation between $A$ and some $X_i$. E.g. Race and Zip Code. 2. It would be beneficial to directly state these assumptions.
The threat model is not clear. From what I gather, privacy is not guaranteed for $X$ only for $A$. This departs from usual formulations and warrants discussion.
3. Setting exact fairness as the goal could be better motivated. The authors follow previous work in this set of assumptions. But since the privacy mechanism is randomized anyways, investigating approximate notions of fairness seems natural. Another issue is for certain distributions $\theta^*$ simply does not exist since the optimization problem is subject to exact fairness constraints.


**Summary Of The Paper:**

This paper characterizes the cost of privacy on exact notions of fairness; exact demographic parity and exact equality of opportunity. The authors use asymptotic relative efficiency.

**Summary Of The Review:**

The paper extends previous work and provides theoretical insight into the cost of sensitive attribute differential privacy. While I did not find anything technically wrong with the paper, the presentation of the paper can be significantly improved with more clarity, better motivation, and more discussion. It could be that there was just not enough space to fully motivate and describe the authors' ideas, I guess I am not sure that ICLR is the best venue for this work.

---

> ### Author Response · Authors · 2022-11-18
> **Response to Reviewer gE91**
>
> We thank the reviewer for the valuable feedback. We address questions and concerns below.
>
> > Implicit assumption that demographic features $A$ are independent of the $X$. In many real-life datasets, there is almost an exact correlation between $A$ and some $X_i$. E.g. Race and Zip Code. 2. It would be beneficial to directly state these assumptions. The threat model is not clear. From what I gather, privacy is not guaranteed for $X$ only for $A$. This departs from usual formulations and warrants discussion.
>
> We analyze the statistical efficiency cost of privacy under Mozannar et al (2020)’s fair learning with private sensitive attributes framework. The reviewer's concern generates a new research question: what is the cost to statistical efficiency if we construct a noisy constraint using imputed sensitive attributes reconstructed from the features? We concur with the reviewer that this research problem is significant and intriguing; however, it does not fit our current mathematical tools and is, therefore, outside the scope of the present paper.
>
> > Setting exact fairness as the goal could be better motivated. The authors follow previous work in this set of assumptions. But since the privacy mechanism is randomized anyways, investigating approximate notions of fairness seems natural.
>
> If we enforce the same approximate notions of fairness (inequality constraints) to the empirical problem and the population problem, we expect to have the same ARE results, but the population-level solution $\theta^\star$ will be different.
>
> > Another issue is for certain distributions $\theta^\star$ simply does not exist since the optimization problem is subject to exact fairness constraints.
>
> We assume the existence of the population-level solution $\theta^\star$ exists. This is true as long as the population-level feasible set is non-empty.
>
> > Notations introduced through out the paper are not always explained (e.g. does the crossed-out operator mean the true mean? (Definition 2.5 and Theorem 3.1)
>
> The crossed-out operator denotes a $0$-valued term. In Definition 2.5 and Theorem 3.1, $c(\theta^\star)$ has a value of $0$ because $\theta^\star$ solves the population problem and therefore satisfies the population constraint $c(\theta^\star) = 0$.
>
> > It is not clear what definitions the authors introduce for the first time and which definitions are cited from prior work. (e.g. is asymptotic relative efficiency an existing concept or newly introduced by the authors?)
>
> Asymptotic relative efficiency is an existing statistical concept that compares the asymptotic variances of two estimators, assuming that both estimators are asymptotically unbiased. Asymptotic relative efficiency in terms of fairness violation is newly introduced by this work to study the cost of privacy in fair machine learning.

---

### Official Review · Reviewer_iTev · 2022-10-25

**Confidence:** 4
**Correctness:** 3
**Technical Novelty And Significance:** 3
**Empirical Novelty And Significance:** 2
**Recommendation:** 5

**Clarity, Quality, Novelty And Reproducibility:**

For the details of these points, please see my review of "Strength and Weakness". However, to give a high-level answer:

**Clarity:** The overall idea is clear. Notation and terminology may get confusing.

**Quality:** The proofs have good quality. The writing, statements, and numerical experiments can be significantly improved.

**Novelty:** The question asked is very specific and original in my view. Note however that the title may look a little overpromise due to the fact that a significant portion of results relies on the privacy mechanism.

**Reproducibility:** The paper provides a theoretical analysis and there is no reproducibility concern as far as I am concerned.

**Strength And Weaknesses:**

**Strength:** I believe the literature is well summarized. The question asked is clear and very relevant to the research our community is conducting. The proofs look correct and the results are intuitive.

**Weaknesses:**
*I list my major and minor concerns that made me recommend marginally below the acceptance threshold. However, I still think the paper has some novelty, hence I hope that my comments would be found useful by the authors to improve the paper's quality. I am also going to stay active during the discussion period in case the authors have updates or questions regarding my review.*

***Major Weaknesses/Questions***
- My biggest concern is that the quantified "efficiency loss" does not tell much. I do not know how the community can use this result in practice. Despite sound mathematical analysis, I think it is not surprising that the efficiency loss is driven by the privacy budget and the (im-)balance between different sensitive attribute groups. If, for example, the quantified loss would be compared with some alternative to the DP attributes + fairness-constrained optimization setting, then perhaps this would guide decision-makers. But right now, the paper defines several assumptions and then expresses the already-imaginable efficiency loss in terms of the (nationally heavily introduced) problem parameters.
- The numerical experiments are not adding anything new to the paper. It is not clear what this tells the reader.
- The paper is hard to follow, because lots of assumptions and notations are introduced before giving the results. Some sentences are objective/personal or unclear. I will list some of them next.

***Minor Weaknesses/Questions***
- Introduction: The second paragraph is not fitting well after the first one. It is not clear without reading the rest of the paper where DP is used. Mozannar et al. (2020) citation is said to propose some technique, but it is not stated in which context this will be useful.
- Page 1, "interpretable" item: The listed reasons for efficiency loss are already known.
- Page 2: We basically study -> maybe we can avoid 'basically'?
- Section 2: In the beginning $\mathcal{Y} := \{ 0,1\} $ is not defined if I am not wrong.
- Section 2 Suggestion: Maybe discuss what happens when some $A$ can be interpreted from $\mathcal{X}$. There is a big literature on this if I am not wrong. Simply saying "we assume $A$ is not included in $X$" might also work.
- $\hat{Y} \triangleq f(X)$ is re-defined more than once
- After Definition 2.1 and 2.2 the notation $\hat{Y} = h(X)$ is used, but $h(\cdot)$ is never defined before.
- When defining the parametric space $\mathcal{H}$, could you maybe show the domain and range of $f_\theta(\cdot)$? Because later on, you will distinguish the cases where these functions can and cannot access $Z$ (private counterpart of $A$).
- "To keep things simple" -> what things? Could you please be more specific here? Because later on you will extend $|\mathcal{A}| = 2$. Is this assumption here for intuition?
- Equation (2.2) and similar problems onward: The use of $\alpha_n$ is not discussed. This is important in my view -- could the authors cite relevant papers (*e.g.*, why why the ERM setting forces us to use such a slack, whether there is a known result on the effect of this parameter on the consistency, etc.)?
- Section 2.2: "In addition, the sampling mechanism $Q$ requires $Z \perp X, Y | A$". What does this mean?
- When "local"-DP is first mentioned, maybe a clarification of what this means would be great. As the DP component of this paper is light, the readers might lack knowledge of DP, hence may be confused about the local DP terminology.
- The last paragraph of Section 2.2 is redundant -- equation (2.4) is obvious, especially after equation (2.3)
- Page 3, $\tilde{f}$ is used for the first time.
- "A direct corollary of Proposition 2.3 is that (2.1) and (2.5) have exactly the same solution $\theta^\star$" -> I don't agree with this. Proposition 2.3 is about the feasibility of a solution in the fairness constraints, but the solution to the underlying optimization problems might be different. Please correct me if you disagree with me and please also clarify this in the paper as the "uniqueness of the solution" explanation might sound vague.
- Please define $\sqrt{n}$-consistent terminology, and the notations of convergence in probability/distribution.
- Section 2.4 "both of them are reasonable" -> What does reasonable mean?
- Question: Is "constraint function" name for $c$ suitable? Typically constraints are defined with their right-hand sides and maybe here we can instead say something as "(signed) fairness violation"?
- Definition 2.5: $\hat{\theta}$ uses $\sigma^2$ however $\tilde{\theta}$ uses $\tilde{\sigma}^2$
- The last two paragraphs of Section 2 are very hard to understand and follow for me. The sentences are using formal notation but the explanations are informal. How can we formally argue that only how close $\hat{\theta} $ converges to $\theta^\star$ is what drives the efficiency: e.g., we still have an inner product with $\nabla c(\theta^\star)$ and it is not very clear.
- Section 3: Please highlight that this generalization extends to regression as well, since before the reader was restricted to classifiers.
- Before defining (3.1) I think we need to discuss $g$ and $h$ as they are not defined.
- Page 5 overall introduces much notation and is not a fun page to read. Can we somehow compress the definitions? Some ideas and further comments: I think (3.3) should be clear from the context and can be moved to Appendix. Defining $\tilde{c}(\theta)$ right after (3.3) looks a bit unnecessary, this is a definition that we keep repeating many times throughout the paper. The same goes for (3.4) and the optimization problems at the end of page 5 -- these are already known and probably there is no need to add the exact same problems only by changing $A$ to $Z$.
- Page 6: In my view, these assumptions should be explained a bit further. Especially, the sub-Gaussian assumption. Is this something used in the related settings? Maybe some citations would "reassure" the reader? The second assumption is very high-level, and such duality/uniqueness arguments can be ensured with simpler lower-level assumptions. As for the third assumption, maybe "Lagrangian" can be clarified further as "Lagrangian dual" and "Lagrangian function" may both be referred to as "Lagrangian".
- Page 7: "the mechanism $Q$ perturbs ..." these are all defined before and I cannot see any benefit of re-stating these with an additional "K".
- Why do the simulations compare a few $n$? Not much intuition is given in the experiments. Why are we ensuring fairness wrt the 'estimation error's? Could you show the linear regression problem satisfies the assumptions that were necessary for the analysis?
- In the Conclusion section referring back to previous mathematical notation such as $h \equiv 1 $ or providing further analysis and introducing new notation is not very usual in my view and it is hard to digest at the end of the paper.

**Summary Of The Paper:**

This paper concentrates on fair machine learning (ML), specifically studying the "sub-optimality" caused when the design phase of a fair estimator is forbidden to have access to the sensitive attributes which it should protect in order to provide fairness. To this end, the authors take the setting where the design phase instead has access to locally-perturbed $\varepsilon$-differentially private versions of the protected attributes, where privacy is ensured via a standard randomized response mechanism. The authors then define and quantify how the resulting estimator deviates from the estimator that has access to the sensitive attributes in terms of statistical efficiency.

**Summary Of The Review:**

The paper studies a modern and interesting topic, and the analysis provided is thorough. I am not sure about how interesting the findings are, and I am concerned about the current presentation of the materials.

---

> ### Author Response · Authors · 2022-11-19
> **Response to Reviewer iTev [Part 1]**
>
> We thank the reviewer for the valuable feedback. We address questions and concerns below.
>
> > My biggest concern is that the quantified "efficiency loss" does not tell much. I do not know how the community can use this result in practice. Despite sound mathematical analysis, I think it is not surprising that the efficiency loss is driven by the privacy budget and the (im-)balance between different sensitive attribute groups. If, for example, the quantified loss would be compared with some alternative to the DP attributes + fairness-constrained optimization setting, then perhaps this would guide decision-makers. But right now, the paper defines several assumptions and then expresses the already-imaginable efficiency loss in terms of the (nationally heavily introduced) problem parameters.
>
> We concur that it is intuitive that the efficiency loss is driven by the privacy budget and the subgroup imbalance; however, this work makes the intuition concrete by introducing a specific metric to analyze such efficiency loss and providing theoretical results for that. As indicated by the title and abstract, the scope is limited to fairness-constrained learning with private sensitive attributes. In future work, we will attempt to generalize our findings to other contexts.
>
> > Page 2: We basically study -> maybe we can avoid 'basically'?
>
> We use the term "basically" because we later extend our theory to missing sensitive attributes that may not be the result of privacy. However, we think the missing sensitive attributes setup is related to privacy, as some users may choose not to disclose their demographic identities during data collection due to privacy concerns.
>
> > Section 2: In the beginning $\mathcal{Y}:=\\{0,1\\}$ is not defined if I am not wrong.
>
> We define this in the second sentence of the second paragraph of Section 2.
>
> > Section 2 Suggestion: Maybe discuss what happens when some $\mathcal{A}$ can be interpreted from $\mathcal{X}$. There is a big literature on this if I am not wrong. Simply saying "we assume $\mathcal{A}$ is not included in $\mathcal{X}$" might also work.
>
> We assume  $\mathcal{A}$ is not included in $\mathcal{X}$. We analyze the statistical efficiency cost of privacy under Mozannar et al (2020)’s fair learning with private sensitive attributes framework. The reviewer's concern generates a new research question: what is the cost to statistical efficiency if we construct a noisy constraint using imputed sensitive attributes reconstructed from the features? We concur with the reviewer that this research problem is significant and intriguing; however, it does not fit our current mathematical tools and is, therefore, outside the scope of the present paper.
>
> > $\hat{Y} \triangleq f(X)$ is re-defined more than once.
>
> We will replace the notation $\triangleq$ by $=$ in the revision.
>
> > After Definition 2.1 and 2.2 the notation $\hat{Y} \triangleq h(X)$ is used, but $h(X)$ is never defined before.
>
> Here $h$ should be $f$. We will fix the typo in the revision.
>
> > When defining the parametric space $\mathcal{H}$, could you maybe show the domain and range of $f_{\theta}(\cdot)$? Because later on, you will distinguish the cases where these functions can and cannot access $Z$ (private counterpart of $A$).
>
> We define the domain and range of a classifier $f$ in the third sentence of the second paragraph in Section 2. We will restate the domain and range of the parameterized classifier in the revision.
>
> > "To keep things simple" -> what things? Could you please be more specific here? Because later on you will extend $|\mathcal{A}| = 2$. Is this assumption here for intuition?
>
> To keep things simple means to keep math and equations simple. Before we extend $|\mathcal{A}| = 2$ to $|\mathcal{A}| = K$, we choose $|\mathcal{A}| = 2$ for the presentation.
>
> > Equation (2.2) and similar problems onward: The use of $\alpha_n$ is not discussed. This is important in my view -- could the authors cite relevant papers (e.g., why why the ERM setting forces us to use such a slack, whether there is a known result on the effect of this parameter on the consistency, etc.)?
>
> We answer this question from two components. First, why do we use slackness? If we do not use slackness (i.e., enforcing the equality constraint), we are likely to get a trivial classifier which is useless. Second, how does the choice of slackness affect the consistency of estimator. A necessary (not sufficient) condition of consistency is that the slackness term is of order $o(\frac{1}{\sqrt{n}})$.
>
> > Section 2.2: "In addition, the sampling mechanism $Q$ requires $Z\perp X, Y | A$". What does this mean?
>
> This means conditioned on the true sensitive attribute $A$, the noisy/private sensitive attribute $Z$ is independent of $X, Y$.

---

> > ### Author Response · Authors · 2022-11-19
> > **Response to Reviewer iTev [Part 2]**
> >
> > > When "local"-DP is first mentioned, maybe a clarification of what this means would be great. As the DP component of this paper is light, the readers might lack knowledge of DP, hence may be confused about the local DP terminology.
> >
> > The intuition of local-DP is explained in the last sentence of the second paragraph in Section 1. The definition of local-DP is introduced by the equation between (2.3) and (2.4). References for local-DP is cited in Section 2.2.
> >
> > > The last paragraph of Section 2.2 is redundant -- equation (2.4) is obvious, especially after equation (2.3).
> >
> > We introduce $\gamma$ for a better presentation instead of a privacy budget $\epsilon$ in the exponent, and recall the reader that we are discussing two-group case. Moreover, we interpret the meaning of privacy budget in terms of $\gamma$ in case some readers might lack knowledge of DP.
> >
> > > Page 3, $\tilde{f}$ is used for the first time.
> >
> > We only use it to claim that we can construct a classifier $\tilde{f}$ which is a function of $X$ and $Z$ jointly.
> >
> > > "A direct corollary of Proposition 2.3 is that (2.1) and (2.5) have exactly the same solution $\theta^\star$" -> I don't agree with this. Proposition 2.3 is about the feasibility of a solution in the fairness constraints, but the solution to the underlying optimization problems might be different. Please correct me if you disagree with me and please also clarify this in the paper as the "uniqueness of the solution" explanation might sound vague.
> >
> > At the population level, the true problem and the noisy problem has the same objective function to minimize and the same feasible set as ensured by Proposition 2.3; consequently, they have the same solution, assuming uniqueness. Beyond the uniqueness of the solution, our theory is founded on perturbation analysis of KKT points, so our results remain valid for non-convex local minima. That is to say, our results are valid so long as a series of empirical estimators converges to a population-level local minimum.
> >
> > > Please define $\sqrt{n}$-consistent terminology, and the notations of convergence in probability/distribution.
> >
> > The term $\sqrt{n}$-consistent and the notations of convergence in probability/distribution used in this paper are standard ones in statistics.
> >
> > > Section 2.4 "both of them are reasonable" -> What does reasonable mean?
> >
> > "Both of them are reasonable" refers to the sentence right before it. Both of the two estimators are consistent, therefore both are reasonable.
> >
> > > Question: Is "constraint function" name for $c$ suitable? Typically constraints are defined with their right-hand sides and maybe here we can instead say something as "(signed) fairness violation"?
> >
> > The name "constraint function" is standard in optimization literature. Constraint function along with their "right-hand sides" constitutes equality constraints or inequality constraints.
> >
> > > Definition 2.5: $\hat{\theta}$ uses $\sigma^2$ however $\tilde{\theta}$ uses $\tilde{\sigma}^2$.
> >
> > Hat notation is reserved for estimators based on samples with true sensitive attributes, tilde notation is for estimators based on samples with noisy sensitive attributes. Both $\sigma^2$ and $\tilde{\sigma}^2$ are population quantities, which are deterministic and not estimated from samples.
> >
> > > The last two paragraphs of Section 2 are very hard to understand and follow for me. The sentences are using formal notation but the explanations are informal. How can we formally argue that only how close $\hat{\theta}$ converges to $\theta^\star$ is what drives the efficiency: e.g., we still have an inner product with $\nabla c(\theta^\star)$ and it is not very clear.
> >
> > An inner product with $\nabla c(\theta^\star)$ restricts the "closeness" between $\hat{\theta}$ and $\theta^\star$ to the linear subspace $\operatorname{ran}(\nabla c(\theta^\star))$. Although we think the last two paragraphs of Section 2 are self-explanatory, we will replace them by a different discussion on other measurements to examine the efficiency loss in the revision.
> >
> > > Section 3: Please highlight that this generalization extends to regression as well, since before the reader was restricted to classifiers.
> >
> > Our results are applicable to supervised learning as long as the problem formulation involves (empirical) risk minimization with fairness constraints.
> >
> > > Before defining (3.1) I think we need to discuss $g$ and $h$ as they are not defined.
> >
> > Examples for $g$ and $h$ are discussed in the paragraph right after (3.1).

---

> > > ### Author Response · Authors · 2022-11-19
> > > **Response to Reviewer iTev [Part 3]**
> > >
> > > > Page 7: "the mechanism $Q$ perturbs ..." these are all defined before and I cannot see any benefit of re-stating these with an additional "K".
> > >
> > > We deliver the intuition behind the mechanism $Q$, which is a "uniform" manner for injecting noise into the sensitive attributes.
> > >
> > > > Why do the simulations compare a few $n$? Not much intuition is given in the experiments. Why are we ensuring fairness wrt the 'estimation error's?
> > >
> > > We compare a few $n$ because ARE is defined in the large sample regime. The simulation problem considers risk parity as a fairness notion to validate our theory.
> > >
> > > > In the Conclusion section referring back to previous mathematical notation such as $h\equiv 1$ or providing further analysis and introducing new notation is not very usual in my view and it is hard to digest at the end of the paper.
> > >
> > > The first paragraph of Section 5 summarizes the paper, while the second paragraph poses a research question for future work.

---

### Decision · Program_Chairs · 2023-01-20

**Decision:**

Reject

**Justification For Why Not Higher Score:**

No enthusiasm (6+) from any of the reviewers.

**Justification For Why Not Lower Score:**

N/A

**Metareview: Summary, Strengths And Weaknesses:**

This studies the sub-optimality in statistical efficiency caused by a recently proposed technique for fairness constrained estimation with sensitive attributes (Mozannar et al,2020)
In this technique local differential privacy is used to perturb "sensitive" attributes in the design of a "fair" estimator. Fairness here refers to various notions of statistical parity among subgroups defined by sensitive attributes. It is a reasonable question to address although its scope is rather narrow. The primary concern with the results is the conceptual and practical implications of the quantified "efficiency loss" as it requires a number of assumptions and parameters that are hard to interpret. The reviewers have also detailed a number of secondary issues with this work.